# Cran1, member of a new class of OLD family ATPases, functions in cell cycle progression in an archaeon

Yunfeng Yang [1], Shikuan Liang[1], Junfeng Liu[1], Xiaofei Fu[1], Pengju Wu[1], Haodun Li[1], Jinfeng Ni[1], Qunxin She [1], Mart Krupovic [2] & Yulong Shen [1✉]

## Abstract

**Overcoming lysogenization defect (OLD) proteins are diverse ATPase-nucleases functioning in antiphage defense in bacteria. However, the role of these proteins in archaea is currently unknown. We describe a new class of archaeal OLD family ATPases and show that they are apparently not involved in antiviral defense but play an essential role in cell cycle progression. The gene for an OLD family enzyme in *Saccharolobus islandicus* REY15A, named here Cran1 (Cell cycle-related ATPase and nickase 1), cannot be deleted and exhibits cyclic expression patterns at transcriptional and translational levels, with peak expression during the transition from M-G1 to S phase. Cran1 overexpression causes significant growth retardation, cell size enlargement, and increased cellular DNA content. Cran1 displays potent nickase and ATPase activities in vitro, with the nickase activity dependent on the presence of the ATPase domain. Notably, Cran1 copurifies with chromatin-associated proteins, such as Cren7 and a histone deacetylase homolog, suggesting its involvement in chromatin-related activities. Collectively, our results suggest that Cran1 plays an important role in cell cycle progression, revealing a novel function of OLD family proteins.**

**Keywords** Archaea; Sulfolobales; OLD Family Protein; Chromatin; Cell Cycle Progression
**Subject Categories** Cell Cycle; Chromatin, Transcription & Genomics; Evolution & Ecology

## Introduction

Overcoming lysogenization defect (OLD) family proteins are ATP-powered nucleases that are widely distributed in bacteria, archaea, and viruses (Dot et al, 2023). Recent bioinformatic, genetic, and structural studies have clarified the functioning and evolution of this protein family (Akritidou and Thurtle-Schmidt, 2023; Schiltz

et al, 2020; Schiltz et al, 2019). These enzymes contain an N-terminal ATPase domain of the ATP-binding cassette (ABC) superfamily and a C-terminal TOPRIM (Topoisomerase-primase) domain nuclease (Aravind et al, 1998). OLD family proteins have been classified into four classes based on gene neighborhood, biochemical properties, domain organization, and catalytic mechanisms (Dot et al, 2023). Class 1 OLD family enzymes are encoded by standalone genes within variable gene neighborhoods and have been implicated in various virus-host and virus-virus antagonistic interactions. Indeed, the founding member of the family has been discovered in bacteriophage P2, with genetic studies showing that the *old*-bearing phage P2 kills *recB* and *recC*-deficient *Escherichia coli* host strains upon infection and specifically interferes with phage λ replication (Lindahl et al, 1970). By contrast, class 2 OLD ATPases-nucleases are always encoded in tandem with UvrD/PcrA/Rep-like helicases (Doron et al, 2018), with the two enzymes, denoted GajA and GajB, respectively, constituting the recently described 'Gabija' antiviral defense system (Akritidou and Thurtle-Schmidt, 2023; Dot et al, 2023). Class 3 OLD ATPases-nucleases are commonly found in retrons, another defense system centered around a reverse transcriptase (*ret*) gene and a non-coding RNA (Gao et al, 2020; Mestre et al, 2020; Millman et al, 2020). Class 4 OLD enzymes constitute the PARIS defense system, with the ATPase and TOPRIM catalytic domains being encoded separately by adjacent genes, denoted *ariA* and *ariB* (Rousset et al, 2022). Although the properties of the OLD family enzymes have been explored extensively, focusing on the abortive infection of P2 lysogens originally discovered in the 1970s and, recently, on the acting mechanisms of several defense systems (Antine et al, 2024; Cheng et al, 2023; Cheng et al, 2021; Doron et al, 2018; Schiltz et al, 2020; Schiltz et al, 2019; Yang et al, 2024), it remains unclear whether the OLD family enzymes have any non-antiviral functions in vivo. To the best of our knowledge, none of the archaeal OLD ATPases-nucleases has been characterized and reported.

Archaea can be divided into four supergroups or superphyla, namely, Euryarchaeota, TACK superphylum, Asgardarchaeota, and DPANN superphylum (Liu et al, 2021c; Spang et al, 2017). In addition to sharing a common origin of information processing machineries with Eukarya, archaea of the Asgardarchaeota and

[1] CRISPR and Archaea Biology Research Center, State Key Laboratory of Microbial Technology, Microbial Technology Institute, Shandong University, 266237 Qingdao, China. [2] Institut Pasteur, Université Paris Cité, CNRS UMR6047, Cell Biology and Virology of Archaea Unit, Paris, France. ✉E-mail: yulgshen@sdu.edu.cn

TACK superphylum encode many proteins that were initially thought to be specific to eukaryotes (Imachi et al, 2020; Liu et al, 2021c). In particular, species of the order Sulfolobales from the TACK superphylum have a cell cycle resembling that of eukaryotes (Gomez-Raya-Vilanova et al, 2025; Lundgren and Bernander, 2005; Lundgren et al, 2008), including G1 (Gap 1), S (DNA synthesis), G2 (Gap 2), M (chromosome segregation), and D (cell division) phases (Lindås and Bernander, 2013). A newborn Sulfolobales cell undergoes a brief interphase G1 (about 5% of the entire cell cycle) during which the cell grows and prepares for genome replication. The cell then enters the S phase, which accounts for about 30% of the cell cycle, during which the genome is duplicated. Then, the cell undergoes a long G2 phase (about 60%), during which cellular metabolism is heightened and cellular volume increases in preparation for genome segregation in the M phase and subsequent cell division in the D phase. The genome segregation and cell division events occur in rapid succession.

The cell division machinery of Sulfolobales consists of archaea-specific protein CdvA, four eukaryotic ESCRT-III homologs, namely, ESCRT-III (also known as CdvB), ESCRT-III-1 (CdvB1), ESCRT-III-2 (CdvB2), and ESCRT-III-3 (CdvB3), and the AAA$^+$ ATPase Vps4 homolog (CdvC) (Lindås et al, 2008; Liu et al, 2017; Risa et al, 2020; Samson et al, 2008). Although the role of ESCRT machinery in the Sulfolobales cytokinesis has been firmly established, the regulation of the cell cycle progression remains unclear. The expression of the *cdvA* is transcriptionally controlled by a repressor aCcr1 (Li et al, 2023; Yang et al, 2023), whereas the subsequent degradation of CdvB by the proteasome is reported to control the cytokinesis progression (Liu et al, 2025a; Risa et al, 2020). However, how the other cell cycle phases, especially G1 and S, are regulated remains unknown. Interestingly, in contrast to Sulfolobales cells, eukaryotic cells have a relatively long G1 phase, during which the activation of different cyclins and Cdk proteins determines whether the cell will enter the cell cycle or remain in a quiescent state (Matthews et al, 2022). Because eukaryotes are believed to have originated from archaea, elucidating the cell cycle regulation in archaea, including the regulation of the M-G1 and S phases, could provide insights into the origin and evolution of the eukaryotic cell cycle regulation mechanisms.

In this study, we characterize an OLD family protein from *Sa. islandicus* REY15A (order Sulfolobales), one of the few genetically tractable representatives of the TACK superphylum. We show that the gene coding for Cran1 is conserved across Sulfolobales and cannot be deleted, suggesting that it is essential for cell viability. Unexpectedly, Cran1 does not appear to participate in antivirus defense, but instead is involved in cell cycle progression. Cran1 exhibits cyclic expression patterns at both transcriptional and translational levels, and its overexpression impairs cell cycle progression. Interestingly, the ATPase activity, but not the nuclease activity, appears to be critical for this function. Our study expands the understanding of the functional versatility of the OLD family proteins and provides insight into the cell cycle regulation in Sulfolobales.

# Results

## Phylogenetic diversity of OLD family proteins

All currently functionally characterized OLD family enzymes, assigned to four classes, are encoded by bacteria or bacterial viruses and are

implicated in antiviral defense (Dot et al, 2023). To assess the diversity and taxonomic spread of this protein family, we assembled a dataset of OLD family proteins from bacteria and archaea (Dataset EV1) and performed maximum likelihood phylogenetic analysis. Notably, not all four classes of characterized OLD family proteins were monophyletic. In particular, Class 2 enzymes were nested within the diversity of Class 1 proteins, whereas Class 4 proteins, in which the ATPase and nuclease domains are split, were nested within Class 3, as a sister group to subgroup 3b. These observations are consistent with the notion that new defense systems evolve through the shuffling of different modules through recombination (Koonin et al, 2017). Whereas Classes 1 and 2 were distributed in both prokaryotic domains, Classes 3 and 4 were restricted to bacteria (Dataset EV1). Notably, phylogenetic analysis uncovered at least 11 well-supported, previously uncharacterized clades (U1-U11) of OLD family proteins (Fig. 1A). Three of these clades (U1, U2, and U4) included both bacterial and archaeal representatives, five clades (U5, U6, U7, U9, and U10) were dominated by diverse bacterial sequences and three clades (U3, U8, and U11) had nearly exclusively archaeal membership (Dataset EV1). Mixed phyletic patterns (e.g., mixed bacterial and archaeal membership) observed in some of the clades are consistent with the horizontal spread characteristic of defense systems (Rocha and Bikard, 2022). By contrast, clades U8 and U11 were homogeneous in their taxonomic representation, restricted to hyperhalophilic (order Halobacteriales) and thermoacidophilic (order Sulfolobales) archaea, respectively. Notably, the phylogenetic relationship between the OLD family sequences in U11 fully recapitulated the species tree of Sulfolobales (Fig. 1B) (Counts et al, 2021), suggesting conservation and vertical inheritance of the corresponding genes in this archaeal order, without obvious signs of horizontal gene transfer. Intrigued by such atypical conservation of the potential OLD family defense genes in Sulfolobales, we set out to characterize them functionally.

## Sulfolobales OLD proteins have expected domain organization and are catalytically active

Analysis of a representative structural model of the Sulfolobales OLD proteins revealed a domain organization typical of this protein family, with the N-terminal ABC family ATPase and the C-terminal TOPRIM nuclease domains. Akin to most other OLD family ATPases, with a notable exception of Class 1 enzymes, Sulfolobales OLD proteins have an α-helical insertion within the TOPRIM domain (Fig. 2A,B). Multiple sequence alignment of the Sulfolobales OLD homologs has shown that all active site residues in both ATPase and nuclease domains are conserved (Appendix Fig. S1), suggesting that the protein is enzymatically active.

To test the in vitro activities of Sulfolobales OLD experimentally, we focused on an OLD protein SiRe_0086 (Cran1) encoded by *Sa. islandicus* REY15A, a thermoacidophilic archaeon growing optimally at 76–80 °C and pH of 2–3. The Cran1 protein was expressed and purified from *Sa. islandicus* cells (Fig. 2C). Notably, whereas OLD family ATPases typically form dimers, in the gel filtration chromatography analysis, Cran1 eluted at a peak of 15.1 ml (Fig. EV1A), corresponding to a monomeric form (55 kDa).

### Nuclease activity
The TOPRIM domain nucleases usually require divalent metal ions for catalysis (Yang, 2011). Thus, we examined the nuclease activity using supercoiled and linearized forms of plasmid pUC19 as the

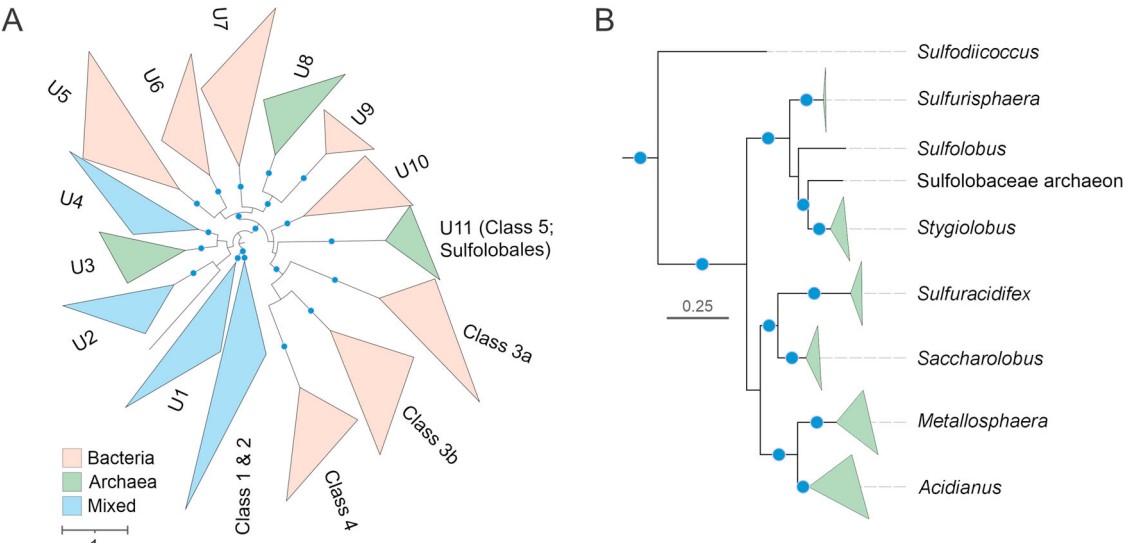

**Figure 1. Diversity of OLD family proteins in bacteria and archaea.**

(A) Maximum likelihood phylogenetic analysis of OLD family proteins. The tree was midpoint rooted. Previously characterized groups of OLD family proteins are denoted by Classes 1 through 4, whereas the clades encompassing uncharacterized protein sequences are labeled U1-U11. Clades that included >90% of proteins encoded by organisms of the same domain were considered as bacterial or archaeal, respectively; when the clades included >10% of proteins encoded by organisms from the other domain, they were denoted as mixed. Amino acid sequences used in the analysis are listed in Dataset EV1. (B) Detailed view of the U11 clade, including OLD family proteins conserved in Sulfolobales. In both panels, well-supported clades were collapsed to triangles, whose side lengths are proportional to the distances between closest and farthest leaf nodes. The scale bars represent the number of substitutions per site, whereas blue circles denote SH-aLRT (Shimodaira–Hasegawa approximate likelihood ratio test) branch supports higher than 90.

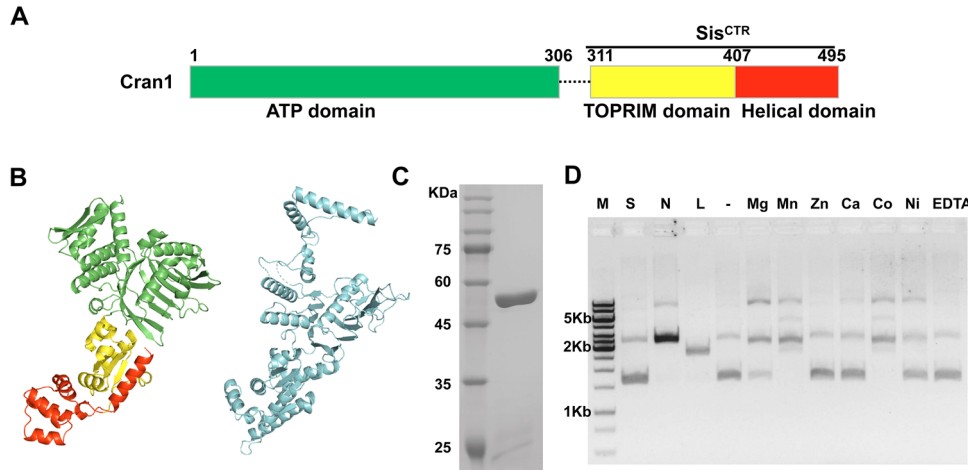

**Figure 2. Cran1 exhibits metal-dependent nuclease activity in vitro.**

(A) Domain architecture of Cran1. Sis$^{CTR}$, the C-terminal domain of Cran1. (B) Left, structural model of Cran1 predicted by Alphafold3 (ipTM = 0.96). The domains are colored as in (A). Right, structure of the Class 1 OLD family protein from *Thermus scotoductus* (blue, PDB:6p74_A). (C) SDS-PAGE showing purified full-length Cran1. (D) Metal-dependent nuclease activity of Cran1 on supercoiled pUC19. "N", "L", and "S" denote the positions of "nicked", "linearized", and "supercoiled" pUC19 plasmid, respectively. "M", DNA marker. '−' no divalent metal ions added. Ethylene diamine tetraacetic acid (EDTA) was added to chelate and deplete the divalent metal ions. Source data are available online for this figure.

substrates in a buffer containing different divalent metal ions (Fig. 2D). In the presence of 10 mM $Mn^{2+}$ and $Co^{2+}$, Cran1 exhibited strong nickase activity, cleaving nearly 100% of the substrate within 20 min (Fig. 2D). In the presence $Mg^{2+}$, ~70% of the supercoiled pUC19 plasmid substrate was nicked within

20 min. By contrast, very weak activity was detected in the presence of $Ni^{2+}$ and no activity was observed with $Ca^{2+}$ and $Zn^{2+}$ as the cofactor (Fig. 2D). These data indicate that specific divalent cations are required for Cran1 activity. Notably, $Ca^{2+}$ alone does not promote nuclease activity on supercoiled DNA substrates; however,

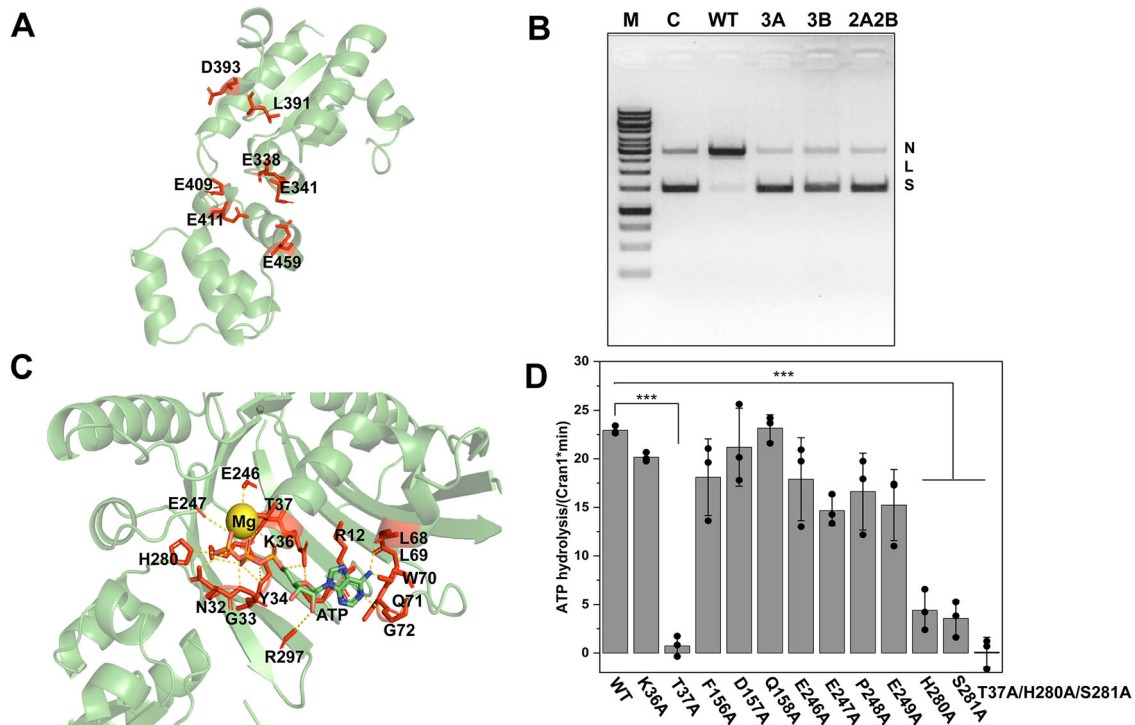

**Figure 3. Cran1 nuclease active site and ATPase key site mutants.**

(A) Metal ion binding residues in the structural domains of nucleases predicted based on protein structure and sequence. (B) Analysis of nickase activity of key residue combination mutants 3A, 3B, 2A2B of nuclease in vitro. WT, the wild-type Cran1 protein. 3A, multiple mutant E338A/L391A/D393A, 3B, multiple mutant E341A/E409A/E411A, and 2A2B, multiple mutant L391A/D393A/E409A/E411A. "N", "L", and "S" denote the positions of "nicked", "linearized", and "supercoiled" DNA, respectively. (C) Structural model illustrating the critical amino acid residues (highlighted in red) located within a 5 Å radius from the active site of the Cran1 ATPase. The structure was predicted by AlphaFold3. (D) Analysis of ATP hydrolytic activity of the mutants of key amino acids of ATPase in vitro. Data from three independent replicates were analyzed by One-sample $t$ test and shown as mean ± SD. ***$P < 0.0005$. Error bars represent standard deviations from three independent measurements. Source data are available online for this figure. Source data are available online for this figure.

with a combination of $Ca^{2+}$ and $Mg^{2+}$, the activity of Cran1 was higher than with either of these ions alone (Fig. EV1B). The enzyme showed the strongest nuclease activity in the presence of 10 mM $Mg^{2+}$ and 1 mM $Ca^{2+}$. Therefore, this combination of divalent cations was used in the subsequent experiments.

Cran1 showed strong nickase activity at a broad range of temperatures (between 40 and 70 °C; Fig. EV1C) and pH (pH 5–10, optimum at pH 6–8; Fig. EV1D). Notably, neither ATP nor its derivatives (ADP, dATP, and AMP) had a measurable effect on the nuclease activity of Cran1 when added to the reaction mixtures (Fig. EV1E). However, the mutants in which the ATPase domain was deleted or the magnesium ion binding/ATP hydrolysis site was mutated lost the nuclease activity (see below), suggesting that an active ATPase domain is necessary for Cran1 to function as a nuclease. Notably, the nuclease activity of Cran1 was inhibited by high salt concentrations (Fig. EV1F).

Our results indicate that Cran1 has similar enzymatic properties to the reported Class 2 OLD family proteins. Based on the previous reports on the mutagenesis of a Class 2 OLD family nuclease (Schiltz et al, 2019) and our structural modeling, we hypothesized that Cran1 uses a two-metal ion catalytic mechanism and predicted that E338, L391, D393, and E341, E409, and E411 are responsible for the binding of metal ions A and B (Fig. 3A), respectively. To test this hypothesis, we constructed three mutants, 3A (E338A/L391A/

D393A), 3B (E341A/E409A/E411A), 2A2B (L391A/D393A/E409A/ E411A), and purified the corresponding proteins (Fig. EV2A). The three mutants completely lost the nuclease activity on the pUC19 plasmid (Fig. 3B), suggesting that both metal-binding sites are essential for the nuclease activity.

**ATPase activity**

Although all OLD family proteins have a typical ATPase domain, not all OLD proteins have ATPase activity (e.g., GajA) (Cheng et al, 2021). To determine whether Cran1 is capable of hydrolyzing ATP, we expressed and purified Cran1 from *Sa. islandicus* and examined its ATPase activity in vitro. As shown in Fig. EV2C,D, Cran1 has ATPase activity with an optimal activity at 70–80 °C. We predicted the key amino acid residues for the ATPase activity by analyzing the structural model generated using AlphaFold3 and sequence alignment (Fig. 3C; Appendix Fig. S1). The proteins in which the conserved residues were individually mutated to alanine were expressed and purified (Fig. EV2B), and their ability to hydrolyze ATP was examined (Fig. 3D). The ATPase activity of T37A, H280A, and S281A mutants was severely impaired, with T37A being the least active, whereas the triple mutant T37A/H280A/S281A completely lost the ATPase activity (Fig. 3D). Notably, the latter mutant also lost the nuclease activity, suggesting a cross-talk between the ATPase and nuclease domains. Consistently, deletion

of the ATPase domain rendered the nuclease domain inactive (Fig. EV2E). Collectively, these results indicate that Cran1 is a functional representative of the OLD family, endowed with both nuclease and ATPase activities. Thus, we denote the clade U11 conserved across Sulfolobales as class 5 (Fig. 1).

## Sulfolobales OLD family proteins do not appear to be involved in antivirus defense

Bacterial class 2–4 OLD enzymes are encoded within conserved genomic loci, including other components of the corresponding defense systems. Genomic neighborhood analysis of the Sulfolobales *old* genes shows that they form a conserved potential operon with a gene encoding a predicted membrane protein of unknown function (*sire_0087* in *Sa. islandicus*; Appendix Fig. S2), further distinguishing the representatives of the class 5 OLD enzymes from the four previously defined OLD classes (Akritidou and Thurtle-Schmidt, 2023; Cheng et al, 2021; Huo et al, 2024; Li et al, 2024; Oh et al, 2023). To study whether Sulfolobales OLD-family proteins are implicated in antivirus defense, we overexpressed Cran1 alone or in combination with the putative accompanying membrane protein SiRe_0087 in *Sa. islandicus* E233S and challenged the recombinant cells with Sulfolobus tenchongensis spindle-shaped virus 2 (STSV2) and Sulfolobus monocaudavirus 1 (SMV1) using a spot assay. Neither of the Cran1 overexpression strains showed detectable antiviral ability. On the contrary, the strain overexpressing Cran1 alone or in combination with the membrane protein SiRe_0087 was more sensitive to STSV2 infection than the control strain (Fig. EV3A). Similarly, overexpression of Cran1 alone or together with SiRe_0087 in *Escherichia coli* MG1655 did not provide detectable protection against bacteriophages T1, T2, T3, T4, T5, and T7 under the assay conditions tested (Fig. EV3B). Although we cannot exclude the possibility that the antiviral activity of Cran1 is specific towards particular viruses (which were not among those tested in this work), our results suggest that Cran1 does not function in antiviral defense but may have other functions.

## Cell cycle-dependent expression of Cran1 appears to be essential for cell viability

We have previously observed that overexpression of aCcr1, a transcription factor that regulates cell cycle progression in *Sa. islandicus* REY15A by suppressing the expression of a gene encoding a key cytokinesis protein CdvA, resulted in fourfold transcriptional repression of *cran1* (Yang et al, 2023). Analysis of the promoter region of *cran1* revealed the presence of the aCcr1 recognition sequence (Yang et al, 2023). Electrophoretic mobility shift assays showed that purified aCcr1 binds to the oligonucleotides encompassing the promoter region of *cran1* (P$_{cran1}$; Appendix Fig. S3), confirming that aCcr1 regulates the transcriptional expression of Cran1.

To analyze whether Cran1 exhibits cell cycle-dependent expression at the protein level, we constructed a *Sa. islandicus* REY15A strain in which the Cran1 is expressed in situ with a Flag-tag at the C-terminus (Appendix Fig. S4A). We verified that the growth and cell morphology of the in situ Flag-tagged Cran1 strain are not different compared to those of the wild-type (Appendix Fig. S4B,C). Then, the cell cycle of the in situ Flag-tagged strain was synchronized with 6 mM acetic acid (Fig. 4A), and the expression

levels of Cran1 at different time points were examined by Western blotting with anti-Flag antibody. As shown in Fig. 4B,C, Cran1 exhibited a cell cycle-dependent expression pattern, with the protein levels being the highest during the M-G1 and S cell cycle phases (i.e., 3–6 h following the removal of acetic acid; Fig. 4BC). The cyclic expression of *Cran1* at the transcriptional level was further confirmed by RT-qPCR using synchronized *Sa. islandicus* E233S cells (Fig. 4D), fully recuperating our previous transcriptomic result (Yang et al, 2023). Notably, during the logarithmic growth phase, the amount of Cran1 in the cells was about five times higher than during the stationary phase (Fig. 4E).

To assess the importance of Cran1 for the cell, we attempted to knock out *Cran1* using the endogenous CRISPR-based genome editing method in *Sa. islandicus* REY15A (Li et al, 2016; Wei and Li, 2023). However, all attempts (at least five times) failed to yield a viable knockout clone, suggesting that *Cran1* is an essential gene. The apparent essentiality of Cran1 contrasts the dispensability of defense systems, including those that have OLD family components in bacteria, and suggests that Cran1 plays an important role for the survival of the cell even in the absence of virus infection. Notably, unlike *cran1*, the gene (*sire_0087*), which forms a putative operon with *cran1* (Appendix Fig. S2) and encodes a hypothetical membrane protein could be knocked out, and the knockout strains exhibited growth and morphology phenotypes similar to those of the wild-type (Appendix Fig. S5). However, our attempts to knock out the whole operon were unsuccessful, further suggesting that the role of Cran1 is different from that of bacterial OLD proteins.

To further assess the importance of Cran1 for the cell, we successfully constructed a *Cran1* knockdown strain using the endogenous CRISPR-Cas-based method (Peng et al, 2015; Zink et al, 2019). As shown in Fig. 5AB, the knockdown strain with four spacers targeting *cran1* had reduced protein level (around 50%) and exhibited apparent growth retardation. Microscopy analysis revealed that the Cran1 knockdown strain generated more (5.6%, *n* = 504) large cells (~2 µm in diameter) when compared to the control strain (0.55%, *n* = 720) (Fig. 5C). Consistent with the growth retardance phenotype, flow cytometry analysis showed that there was slight difference, although not significant, between the knockdown and the control strains, with smaller population of 1C cells in the knockdown strain than in the wild-type strain, and population of cells exceeding 2C was bigger than that in the wild-type strain (Fig. 5D), suggesting that chromosome segregation and/or cell division was inhibited and some 2C cells entered into the S phase without cell division. The relatively mild phenotype obtained with the knockdown strain may be linked to the limited efficiency of the transcript depletion, i.e., cells with 50% of the native Cran1 may function normally. Nevertheless, these results further suggest that Cran1 plays an important role in *Sa. islandicus* cell cycle.

## Overexpression of Cran1 variants with the wild-type ATPase domain impairs cell cycle progression

Strikingly, overexpression of Cran1 resulted in dramatic growth retardation (Fig. 6A), which was associated with a considerable increase in cell size, most noticeably after 24 h post-induction (average diameter of 4.54 µm; *n* = 100) compared to the control cells (~1.2 µm), with 95% of the cells exhibiting 50-fold increase in volume (Fig. 6B,C). In addition, flow cytometry analysis showed an increase in the number of genome equivalents within the

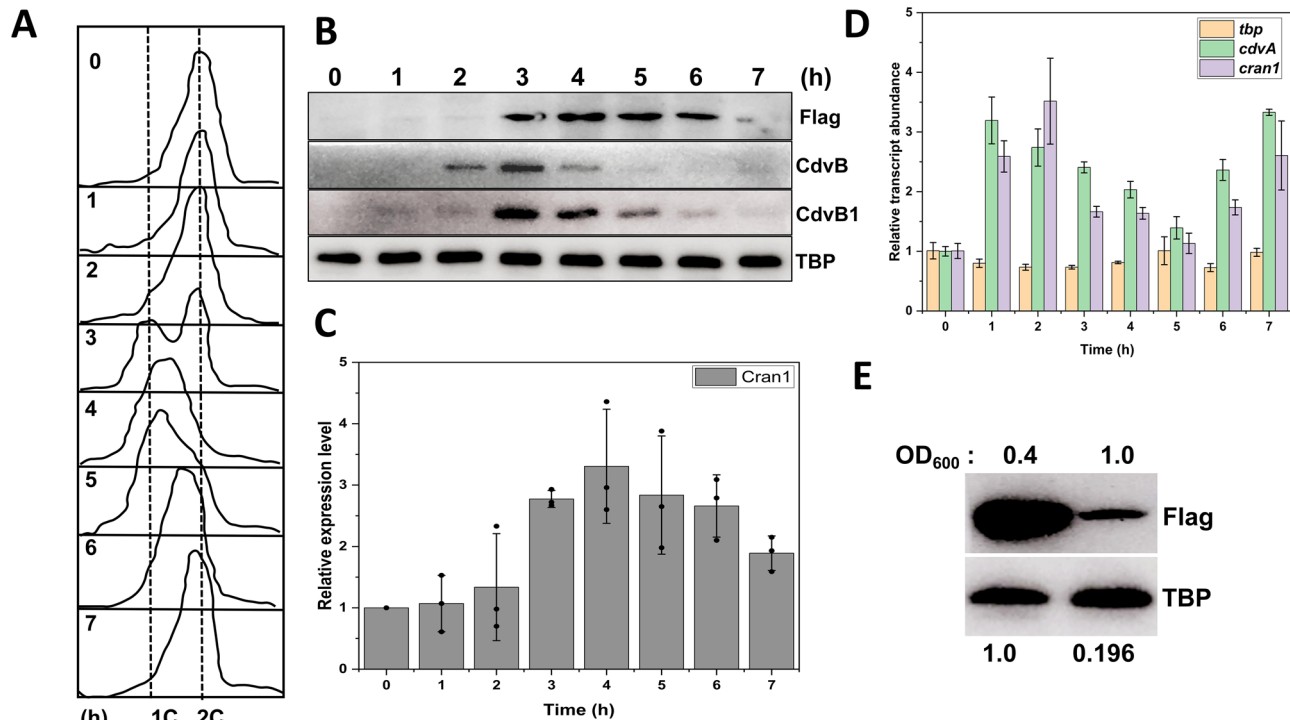

**Figure 4. The cyclically expressed gene *cran1* is essential for cell viability.**

(A) Flow cytometry profiles of samples of a synchronized Cran1 in situ-flag strain in one cell cycle. The cells were synchronized at G2 phase after being treated with 6 mM acetic acid for 6 h before being released by removing the acetic acid. The cultures were collected at different time points (0, 1.0, 2.0, 3.0, 4.0, 5.0, 6.0, and 7.0 h) and subjected to flow cytometry analysis. Cells started to divide at 2.0 h as shown by the appearance of cells with one copy of a chromosome (1C), and the ratio of dividing cells reached the highest level at about 3–3.5 h. As the cell cycle proceeded, cells with two copies of chromosomes (2 C) became dominant at 7 h. (B) Changes of the protein levels of Cran1 during a cell cycle detected by western blotting using anti-Flag antibody. The known periodic expression of CdvB and CdvB1 were used as positive controls, and TBP (TATA-box binding protein) was used as a loading control. (C) Quantification of the result in (B). The values were based on three biological replicates. Error bars represent standard deviations from three independent measurements. (D) Detection of transcriptional levels of *cdvA* and *cran1* during a cell cycle RT-qPCR. The values were obtained based on three biological replicates. Error bars represent standard deviations from three independent measurements. (E) Western blotting to the protein levels of Cran1 during exponential (OD$_{600}$ = 0.4) and stationary (OD$_{600}$ = 1.0) phases, respectively. Source data are available online for this figure.

population of large cells (Fig. 6D). All these phenotypes imply that Cran1 is involved in cell cycle progression. By contrast, overexpression of SiRe_0087 had no effect on the growth and cell morphology (Appendix Fig. S5B–D). Co-expression of Cran1 and SiRe_0087 yielded phenotypes generally similar to expression of Cran1 alone (Appendix Fig. S5B–D). These results suggest that although *sire_0087* and *cran1* are putatively located in the same operon, their functions may not be directly linked or correlated. To analyze whether the cell cycle-related function of Cran1 is conserved in other members of the order Sulfolobales, we overexpressed a Cran1 ortholog from *Sulfolobus acidocaldarius* DSM639 (SacCran1) in *Sa. islandicus* E233S. SacCran1 overexpression also resulted in growth retardation, larger cell size, and increased cellular DNA content (Fig. EV4), suggesting that the cell cycle-related function of Cran1 is conserved in Sulfolobales, consistent with the conservation of gene neighborhood of *cran1* in other members of the order (Fig. 1B; Appendix Fig. S2).

In order to explore whether the enzymatic activities of Cran1 are important for the observed effects, we constructed *Sa. islandicus* strains overexpressing different ATPase active site mutants and tested their phenotypes. Overall, the effects of different mutants on the culture growth rates and cell morphology were consistent with the in vitro ATPase activity results. In particular, the strains

expressing the ATPase mutants that had the strongest impacts on the ATP hydrolysis in vitro approached the phenotypes of control cells containing the empty vector, whereas mutants in which the ATPase activity was only mildly affected, resembled the strain overexpressing the wild-type Cran1 and displayed growth retardation and an increase in cell size. In the strain overexpressing Cran1(T37A), a mutant with severely compromised ATPase activity in vitro, the majority of cells (but not all) were of normal size, and their growth rate was the most closest to that of the strain carrying the empty vector pSeSD (Fig. 7A). The strains expressing Cran1(H280A) and Cran1(S281A) also had improved growth rate compared to the cells overexpressing the wild-type Cran1 (Fig. 7A). Notably, the Cran1(S281A) cells were enlarged and more irregular, with bud-like formations and dumbbell-like morphologies (Fig. 7E), compared to the other mutant cells that were uniformly enlarged and spherical. Similar abnormal cell morphologies were observed in the case of the cell division-impaired *Sa. islandicus* strain overexpressing the dominant-negative mutant of the key cytokinetic ATPase, Vps4 (Liu et al, 2025b). It is noteworthy that in the class 1 and 2 OLD proteins, mutation of the key aspartic acid residue in the Walker B motif results in a complete loss of the ATPase activity. In Cran1, the Walker B motif residues E246 and E247 are predicted to coordinate a magnesium ion. However, in our

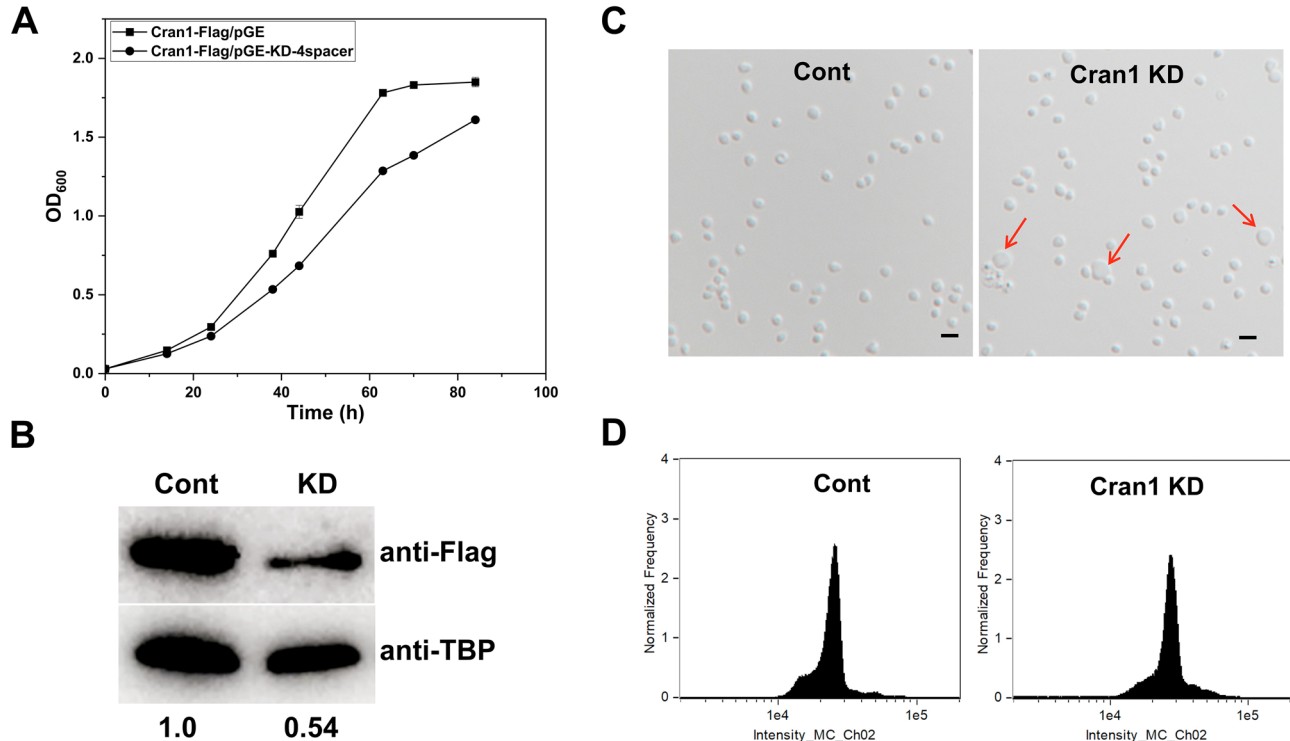

**Figure 5. Phenotypic analysis of Cran1 knockdown strains.**

(A) Based on the knockdown technology of endogenous type III CRISPR in *Sa. islandicus*, an interfering plasmid possessing four tandem repeat spacers, was introduced into the Cran1 in situ plus Flag-tagged strains to obtain Cran1 knockdown strains, and the growth curves were determined. (B) Western blotting was used to detect protein levels in Cran1 knockdown strains, quantified in gray values using ImageJ software. (C) Differential interference contrast (DIC) mode microscopy and (D) flow cytometry of Cran1 knockdown strains. Cells cultured in the induction medium ATV were taken at 24 h and observed under an inverted fluorescence microscope. DNA content of cells was analyzed using an ImageStreamX MarkII Quantitative imaging analysis flow cytometry (Merck Millipore, Germany). Scale bars: 2 μm. The red arrow points to the larger cells produced by the Cran1 knock-down strain. Source data are available online for this figure.

experiments, substitution of either of the two residues with alanine had no effect both in vitro and in vivo. Analysis of the structural model suggested that T37 is closer to the magnesium ion than E246 and E247, and may play a more important role in magnesium ion coordination or in maintaining a functional conformation. This may explain why mutation of either Walker B residue had no significant effect on the ATPase activity (Fig. 3D) but still led to growth retardancy (Fig. 7B).

Given that none of the single mutants completely recovered the phenotype of control cells, we constructed a strain overexpressing Cran1(T37A/H280/S281A), a triple mutant, which lost both the ATPase and nuclease activities in vitro (Fig. 3D). Overexpression of this mutant in vivo yielded cells with morphology identical to that of the control cells (Fig. 7B,E). Unexpectedly, *Sa. islandicus* strains carrying plasmids overexpressing any of the nuclease active site mutants, which abrogate the nuclease activity in vitro, showed the same phenotypes as those observed in the strain overexpressing the wild-type Cran1 (Fig. 7C,D). We verified that the proteins were expressed at similar levels in all the mutant strains by western blotting, confirming that the phenotypic difference was not caused by differences in the expression levels (Fig. EV2F). Collectively, our results suggest that an intact ATPase domain is required for the Cran1 variants to interfere with the normal progression of the cell cycle.

## The overexpression of Cran1 results in arrest at the S phase without undergoing cell division

To analyze which of the cell cycle phases was impacted by the overexpression of Cran1, we synchronized the Cran1 and Cran1(T37A/H280A/S281A) overexpression strains to G2 phase, with cells harboring the empty vector pSeSD used as a control (Fig. EV5A). To ensure sufficient protein concentration, arabinose was added to induce protein expression two hours before acetic acid removal. The control cells started to divide ~4 h following the release of the cell cycle arrest. Profiles similar to those of the control cells were obtained for the strain overexpressing the inactive Cran1 ATPase mutant Cran1(T37A/H280A/S281A) (Fig. EV5B). By contrast, the division of the wild-type Cran1 overexpression strain was blocked. Notably, although the cell cycle could not progress into the cell division phase, genome replication continued in the cells expressing the wild-type Cran1 leading to the accumulation of intracellular DNA, suggesting that the cells were held at the S phase incapable of cell division.

## Cran1 may be involved in chromatin processing-related activities in M-G1 and S phases

To gain further insights into how Cran1 functions, we determined the subcellular localization of Cran1 by chromatin fractionation. As

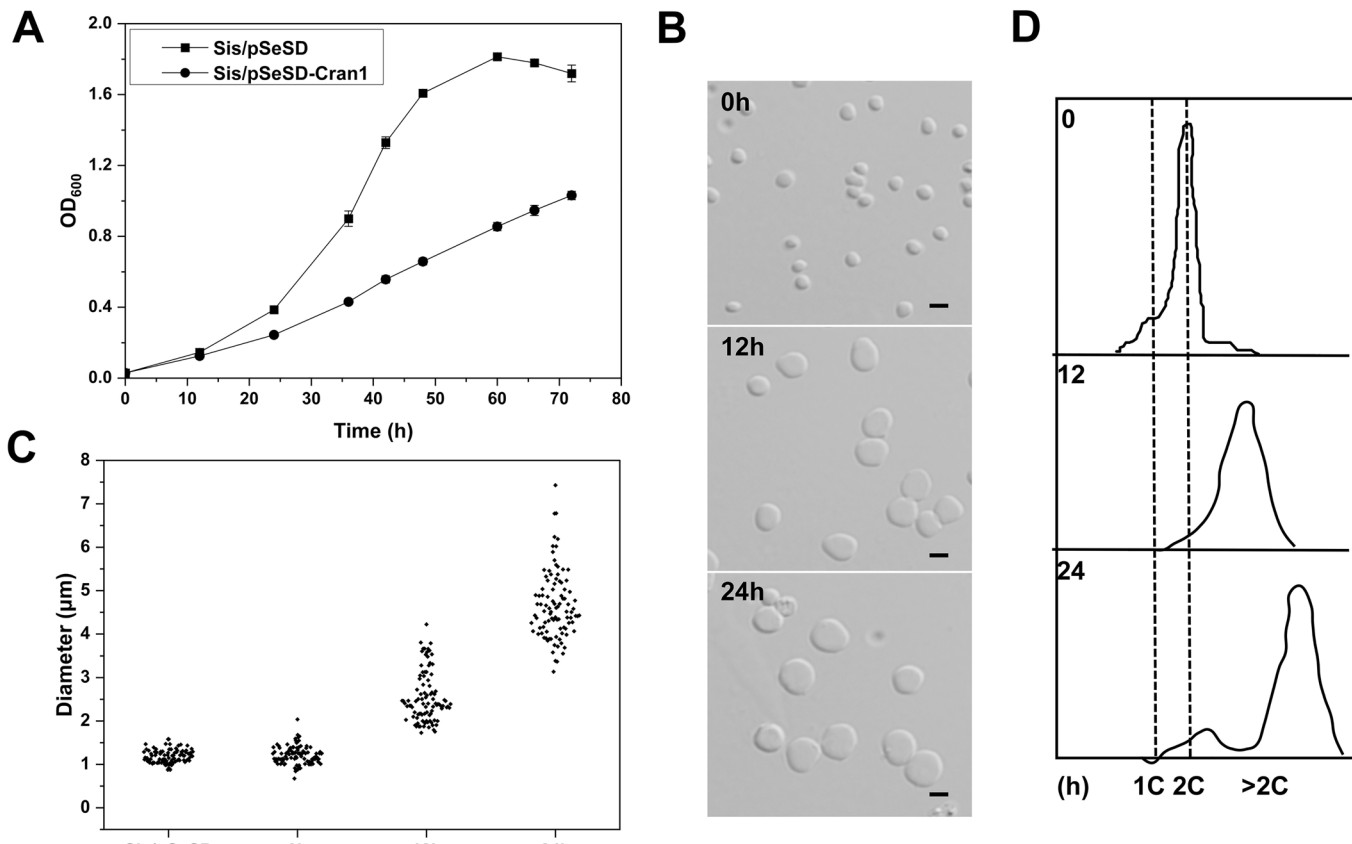

**Figure 6. Overexpression of Cran1 leads to retarded cell growth and cell enlargement.**

(A) Growth curves of cells overexpressing the C-terminal His-tagged Cran1. The cells were inoculated into 30 ml induction medium ATV to a final estimated $OD_{600}$ of 0.03. The growth was monitored at $OD_{600}$. Each value was based on data from three independent measurements. Error bars represent standard deviations from three independent measurements. Cells harboring the empty plasmid pSeSD were used as a control. (B) Bright-field microscopy (DIC) of cells overexpressing Cran1. Cells cultured in the induction medium were taken at different time points and observed under a NIKON-E microscope. Upper, middle, and lower panels, 0, 12, and 24 h. Scale bars: 2 µm. (C) Cell size statistics of the data in (B). Cell cultures were sampled at the indicated time points and observed under the microscope. The diameters of ~100 cells were measured using ImageJ software for each culture repeat. (D) Flow cytometry analysis of the DNA content of Sis/pSeSD and Sis/pSeSD-Cran1 cells cultured in MATV medium. Source data are available online for this figure.

shown in Fig. 8A, a fraction of Cran1 was associated with the chromatin in the logarithmic growth phase, whereas almost all of the protein was located in the cytoplasm during the stationary phase (Fig. 8A). To assess whether Cran1 is involved in chromatin-associated activities, we performed pulldown assay using genome coded Flag-tagged Cran1 as a bait, followed by identification of the purified proteins by mass spectrometry. Interestingly, multiple chromatin-associated proteins co-purified with Cran1, with Cren7 and histone deacetylase superfamily protein (SiRe_1988) being the most abundant (Fig. 8B,C; Dataset EV2). Notably, both Cran1 and Cren7 are among the essential genes regulated by aCcr1 (Yang et al, 2023). These results suggest that Cran1 likely functions in cooperation with chromatin proteins.

## Discussion

Here, we characterized a representative of a new class of OLD family proteins, class 5, and the first OLD family protein from archaea. Contrary to our initial expectation, Cran1 did not appear to participate

in the antivirus defense but was rather involved in the cell cycle progression. Consistently, unlike in the case of bacterial defense-related OLD proteins, which are not essential for the normal cell growth or viability (Akritidou and Thurtle-Schmidt, 2023), the *cran1* gene of *Sa. islandicus* could not be deleted. Cran1 displays cyclic expression, with the protein level being high during the progression of the cell cycle from the M-G1 into the DNA synthesis (S) phase, and being depleted during the post-replicative G2 phase (Fig. 4B). Importantly, Cran1 is enriched in the chromatin fraction in vigorously growing cells and immunoprecipitation showed that Cran1 is associated with the chromatin protein Cren7 and a histone deacetylase homolog. Overexpression of Cran1 does not affect genome replication but appears to impair chromosome segregation and/or cell division. Collectively, these observations suggest that Cran1 functions during the M-G1 and S phases and is involved in chromatin processing-related activities, although the possibility that it plays a role in antiviral defense under particular circumstances or against specific viruses cannot be completely excluded.

The knockdown of *cran1* resulted in only 50% depletion of the expression. Although the strain displayed mild but notable changes

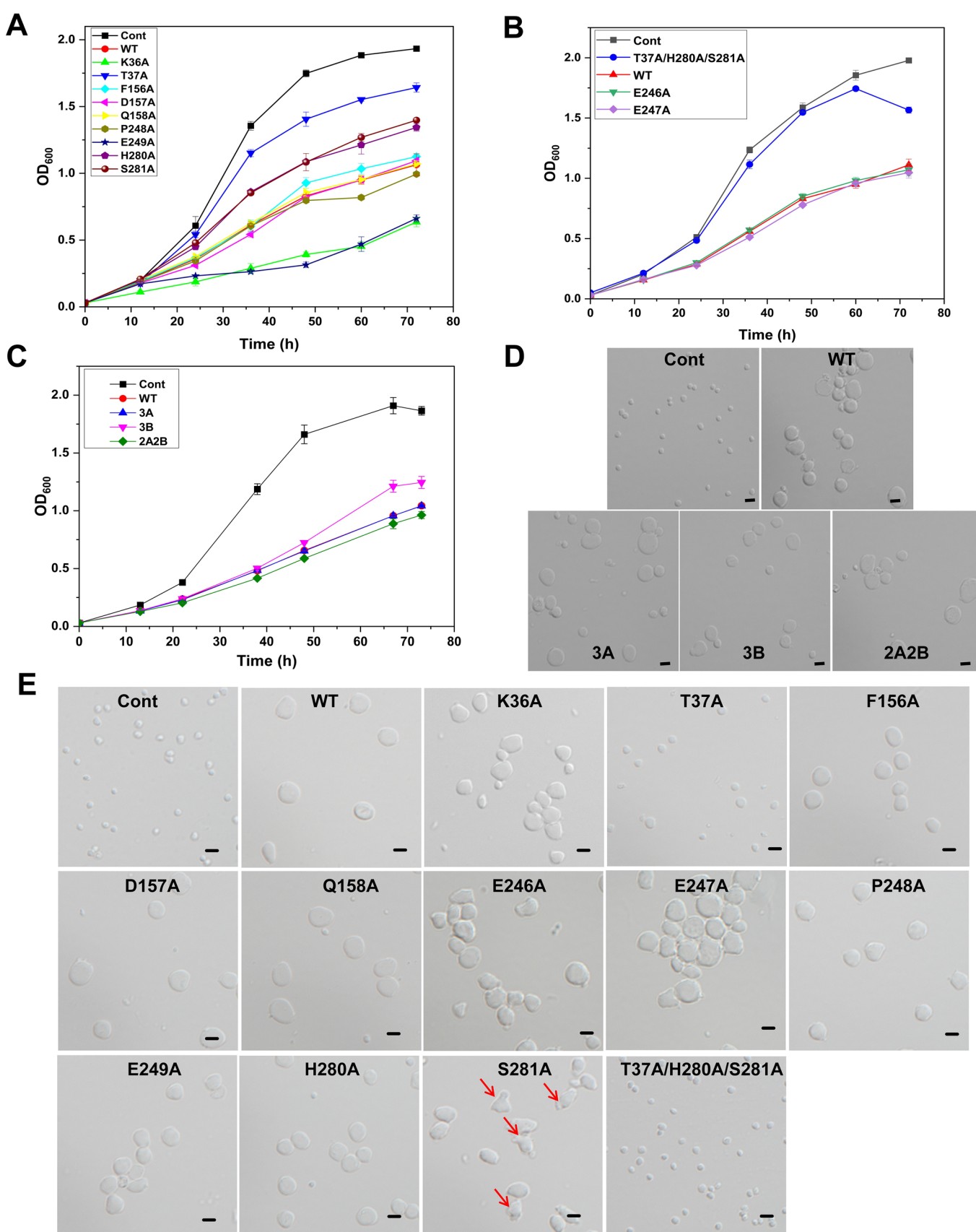

**Figure 7. The ATPase activity of Cran1 is vital for cell cycle progression.**

(A) Growth curves of the predicted single key amino acid residue mutant overexpression strains. Data from three independent replicates are shown as mean ± SD. Source data are available online for this figure. Error bars represent standard deviations from three independent measurements. (B) Growth curves of the strains overexpressing mutants Cran1(E246A), Cran1(E247A), and Cran1(T37A/H280A/S281A). Data from three independent replicates are shown as mean ± SD. Source data are available online for this figure. Error bars represent standard deviations from three independent measurements. (C) Growth curves of the putative metal ion binding site mutant overexpression strains for the nickase activity. "3A" denotes E338A/L391A/D393A; "3B", E341A/E409A/E411A; "2A2B", the L391A/D393A/E409A/E411A. Data from three independent replicates are shown as mean ± SD. Source data are available online for this figure. Error bars represent standard deviations from three independent measurements. (D, E) Representative images of cells of the nickase-deficient mutant overexpression strains in (C) and the predicted key amino acid residue mutant overexpression strains for the ATPase activity in (A, B), respectively. Samples were taken from the arabinose induction for 24 h. Scale bars, 2 μm. The red arrow points to irregularly shaped cells produced by overexpression of the S281A mutant. Source data are available online for this figure.

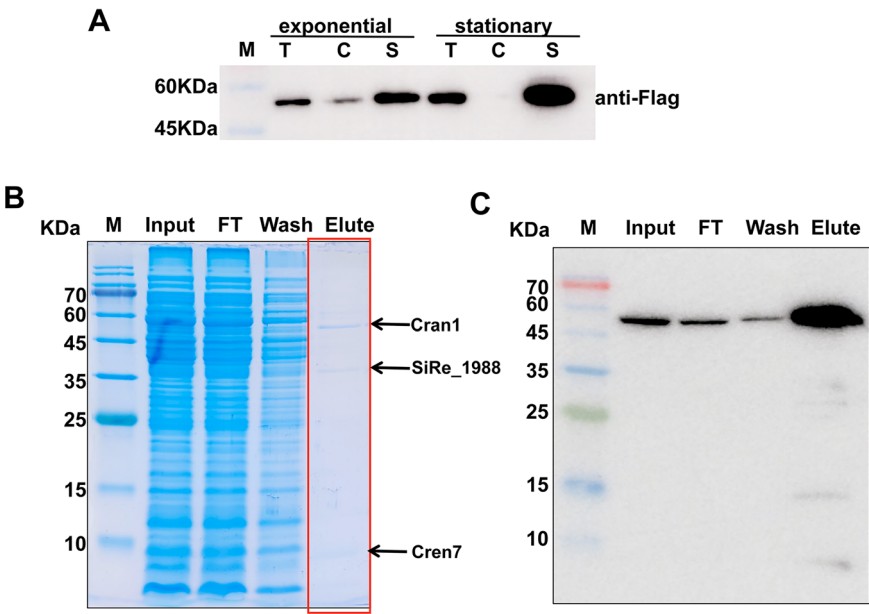

**Figure 8. Subcellular localization of Cran1.**

(A) Cellular localization of Cran1 in the cytoplasm or chromatin of cells during the exponential phase (OD$_{600}$ = 0.4) of cell growth as well as in stationary cells (OD$_{600}$ = 1.0). T total protein, C chromatin fractions, S soluble fractions. (B) SDS-PAGE and (C) western blotting of immunoprecipitation of Cran1 in situ Flag strains. Input represents cell lysate, FT indicates flow through, Wash indicates washing of magnetic beads using buffer A to remove nonspecifically bound fractions, and Elute indicates elution fractions of target proteins. Source data are available online for this figure.

in the growth and cell division phenotypes, the cells were still able to complete the cell cycle progression. By contrast, the cell cycle was strongly impaired in the Cran1 overexpression strain, with the cells displaying an increase in size and chromosomal copy numbers, suggesting blockage of cytokinesis. We hypothesize that during the normal cell cycle progression, Cran1 levels need to be reduced during the G2 phase prior to cell division (Fig. 4B), when chromosomes undergo condensation in preparation for segregation. This is probably achieved through a combination of transcriptional regulation by aCcr1 and degradation of the protein (Risa et al, 2020; Yang et al, 2023). It has been reported that the half-life of the *SSO2277* mRNA, a close ortholog of Cran1 from *Sa. solfataricus*, is only 3 min (Andersson et al, 2006). Furthermore, there was a slight accumulation of SacCran1 (Cran1 from *Sulfolobus acidocaldarious*) when the cells were treated with the proteasome inhibitor Bortezomib (Risa et al, 2020). These observations imply that Cran1 is tightly controlled at the RNA level and could be further regulated at the protein level by

proteasome degradation. The dual regulation in vivo emphasizes the importance of the cell cycle phase-dependent function of this protein.

The exact role of Cran1 during cell cycle progression remains unclear. However, given that Cran1 expression is elevated following cell division, during the M-G1 and S phases, we hypothesize that Cran1 facilitates genome replication by relaxing the chromosome using its nickase activity. By contrast, overexpression of Cran1 beyond the S phase would be expected to interfere with chromosome condensation and subsequent cytokinesis. Notably, the inability to undergo cell division does not appear to prevent the subsequent round of genome replication, further suggesting that genome replication and cytokinesis are not strictly coupled in *Sa. islandicus*, consistent with previous results (Liu et al, 2025b). This could explain the emergence of large cells with multiple copies of the genome. This result is consistent with the more efficient proliferation of the STSV2 virus in the Cran1 overexpression strain. Indeed, similar to some other viruses, STSV2 infection blocks cell

division and arrests the cell cycle in the DNA synthesis phase, which ensures its access to the host's replication machinery (Fan et al, 2018; Kiro et al, 2013; Liu et al, 2021a; Stewart et al, 2013). The protein level of Cran1 during the exponential growth phase is five times that during the stationary phase, making it possible for Cran1 to promote genome replication during the rapid growth phase.

In eukaryotes, histone modification is an important mechanism for regulating genome accessibility (Bannister and Kouzarides, 2011). Immunoprecipitation experiments showed that Cran1 is associated with histone deacetylase homolog and Cren7, Sac7d/ Sso7d, SSB (single-stranded DNA binding protein), and Alba (Dataset EV2). Chromatin proteins have various post-translational modifications (acetylation, methylation, phosphorylation, etc.) (Wolffe and Hayes, 1999), which affect not only the properties of the proteins themselves, but also the structure of the chromatin and gene expression. For example, many chromatin proteins of *Sa. islandicus* have been reported to be post-translationally acetylated or methylated (Cao et al, 2019; Vorontsov et al, 2016). Therefore, post-translational modification of the chromatin proteins is likely also to play an important role in cell cycle progression in archaea. Given that Cran1 has ATPase and nickase activities, we hypothesize that it plays a role in chromatin relaxation and/or unwinding of supercoiled DNA in cooperation with chromatin proteins, such as Cren7, during the S phase, facilitating DNA replication and subsequent chromosome segregation. However, the underlying mechanism remains unknown.

Structural and biochemical studies have greatly increased our understanding of the antiphage functions of the OLD family proteins in bacteria in recent years. In the Gabija antiviral system, GajA, a representative of the Class 2 OLD family, forms a tetramer, with the ATPase domain clustered at the center and the Toprim domain located peripherally (Li et al, 2024). ATP binding by the ATPase domain stabilizes the insertion region within the ATPase domain, keeping the Toprim domain in a closed conformation. Upon depletion of ATP by phages, the Toprim domain undergoes conformational changes, allowing it to efficiently bind and cleave DNA substrates (Li et al, 2024). Notably, GajA has likely lost its ATP hydrolysis activity during evolution but retained the nucleotide-binding capacity (Cheng et al, 2021), with the ATPase domain playing a regulatory role, stimulating the nuclease activity in response to viral invasion (Cheng et al, 2023; Cheng et al, 2021). Similarly, the ATPase domains of the Class 2 OLD proteins BpOLD (*Burkholderia pseudomallei* OLD) and XccOLD (*Xanthomonas campestris pv. Campestris* OLD), which catalyze the nicking and cleavage of DNA substrates, are thought primarily to promote oligomerization and control how and when the nuclease domain gains access to the substrate (Schiltz et al, 2019). We found that deletion or inactivation of the Cran1 ATPase domain abolishes the nuclease activity, suggesting that there is a cross-talk between the two domains, similar to what has been reported for other OLD family proteins. Given that the presence or absence of ATP, dATP, ADP, and AMP did not affect the nuclease activity of Cran1 in vitro, we presume that the proteins were co-purified with the bound nucleotide. It is tempting to speculate that it is the inherent cross-talk between the sensor ATPase domain and the effector nuclease domain that predisposed the OLD family enzyme for recruitment as an important player in the cell cycle control in Sulfolobales. Indeed, as evident from the in vivo experiments with

the strain overexpressing the nucleotide-binding and ATPase-deficient mutant (T37A/H280A/S281A), which did not show growth inhibition, the ATPase domain of Cran1 is crucial for its function in vivo. In conclusion, our study has revealed an unexpected novel function of an archaeal OLD family protein in cell cycle progression.

# Methods

**Reagents and tools table**

| Reagent/resource | Reference or source | Identifier or catalog number |
| --- | --- | --- |
| **Experimental models** | | |
| *Sulfolobus islandicus* REY15A | Contursi et al, 2006 | Not applicable |
| *S. islandicus* REY15A (E233S) | Deng et al, 2009 | Not applicable |
| *Escherichia coli* DH5α | Laboratory strain | |
| *E. coli* BL21(DE3) codon plus-RIL | Laboratory strain | |
| *E. coli* MG1655 | From the She lab | ATCC:47076 |
| **Recombinant DNA** | | |
| Plasmids | This study | Appendix Table S1 |
| Primers | This study | Appendix Table S2 |
| **Antibodies** | | |
| Rabbit anti-CdvB | Liu et al, 2025b | |
| Rabbit anti-CdvB1 | Liu et al, 2025b | |
| Rabbit anti-TBP | Liu et al, 2025b | |
| Rabbit anti-His tag | TransGen Biotech | HT501-01 |
| Rabbit anti-Flag tag | TransGen Biotech | HT201-01 |
| Goat anti-Rabbit IgG (H + L)-HRP Conjugate | Kermey Biotech | MAR001 |
| **Oligonucleotides and other sequence-based reagents** | | |
| **Chemicals, enzymes, and other reagents** | | |
| *ApexHF* HS DNA Polymerase FS | Accurate Biotechnology, Hunan, China | AG12202 |
| 3-Color Prestained Protein Standards | Accurate Biotechnology, Hunan, China | AG11919 |
| Evo M-MLV RT Kit with gDNA Clean for qPCR | Accurate Biotechnology, Hunan, China | AG11728 |
| SYBR Green Premix Pro Taq HS qPCR Kit | Accurate Biotechnology, Hunan, China | AG11701 |
| SparkZol Reagent | SparkJade Co., Shandong, China | AC0101-B |
| Sparkjade ECL super Western blotting detection reagents | SparkJade Co., Shandong, China | ED0015-B |
| QuickCut SphI | Takara | Code No.1632 |
| QuickCut XhoI | Takara | Code No.1635 |
| QuickCut NdeI | Takara | Code No.1621 |
| QuickCut SalI | Takara | Code No.1636 |
| QuickCut HindIII | Takara | Code No.1615 |

| Reagent/resource | Reference or source | Identifier or catalog number |
|---|---|---|
| BspQI | Takara | Code No.1227 A |
| T4 DNA Ligase | Takara | Code No.2011A |
| TEDA ligation system | Xia et al, 2019 | |
| Bradford protein assay kit | Beyotime | P0006 |
| Acetic acid | Sigma-Aldrich | A6283-100ML |
| Propidium iodide | Sigma-Aldrich | P4170-25MG |
| Isopropyl-thiogalactopyranoside (IPTG) | Sangon Biotech | A600168-0005 |
| Malachite green | Sangon Biotech | A600591-0050 |
| Ammonium molybdate | Sangon Biotech | A600067-0100 |
| Triton X-100 | Sangon Biotech | A600198-0500 |
| ATP | Sangon Biotech | A425673-0100 |
| dATP | Sangon Biotech | B500044-0001 |
| ADP | Sangon Biotech | A351243-0010 |
| AMP | Sangon Biotech | A425244-0001 |
| Sorbitol | Sangon Biotech | A430707-0200 |
| PMSF | Sigma-Aldrich | P7626 |
| **Software** | | |
| Curve graph drawing | OriginPro 2024 (64-bit) SR1 10.1.0.178 software | Not applicable |
| PyMOL | PyMOL 3.1 software | Not applicable |
| Sequence alignment | DNAMAN V6 software | Not applicable |
| **Other (online resources)** | | |
| NCBI | https://www.ncbi.nlm.nih.gov/ | Not applicable |
| Alphafold3 | https://alphafoldserver.com/ | Not applicable |
| ImageJ | https://ij.imjoy.io/ | Not applicable |
| IQ-Tree | https://iqtree.github.io/ | Not applicable |
| SyntTax | https://archaea.i2bc.paris-saclay.fr/SyntTax/ | Not applicable |

## Strains and growth conditions

*Sa. islandicus* E233S($\Delta pyrEF\Delta lacS$) (Reagents and tools table) was grown in STVU medium containing mineral salt, 0.2% (w/v) sucrose (S), 0.2% (w/v) tryptone (T), 0.01% (w/v) uracil (U), and a mixed vitamin solution (V). The medium was adjusted to pH 3.3 with sulfuric acid, as described previously (Deng et al, 2009). STV medium containing 0.2% (w/v) tryptone (T) was used for screening and cultivating uracil prototrophic transformants. ATV medium containing 0.2% (w/v) D-arabinose(A) was used for protein expression.

Culture plates were prepared using gelrite (0.8% [w/v]) by mixing 1× STV and an equal volume of 1.6% gelrite. Plasmid cloning and protein expression were performed with *E. coli* DH5α and BL21 (DE3)-RIL, respectively. The cells were cultured in Lysogeny-Broth (LB) medium supplemented with appropriate antibiotics. Plaque assay was performed using *E. coli* MG1655. The strains constructed and used in this study are listed in Appendix Table S1.

## Phylogenetic analysis

The phylogenetic analyses were similar to those previously described (Yang et al, 2023). In short, a dataset of OLD family proteins from bacteria and archaea was collected by PSI-BLAST (2 iterations against the RefSeq database at NCBI; $E = 1e-05$) (Altschul et al, 1997) (Dataset EV1). The collected sequences were then clustered using MMseq2 to 90% identity over 80% of the protein length. In the case of Class 4 OLD proteins, the ATPase and nuclease subunits were concatenated. Sequences were aligned using MAFFT v7 (Katoh et al, 2019) and the resultant alignment trimmed using Trimal (Capella-Gutiérrez et al, 2009), with the gap threshold of 0.2. Maximum likelihood phylogenetic analysis was performed using IQ-Tree (Minh et al, 2020), with the best selected amino acid substitution model being LG + I + G4. The branch support was assessed using SH-aLRT (Guindon et al, 2010).

## Construction of plasmids

Construction of *cran1* knockout and in situ gene tagging of *cran1* strains was according to the endogenous CRISPR-Cas-based genome editing method as described previously (Li et al, 2016). A target site was selected on the *cran1* consisting of a 5'-CCN-3' type I-A protospacer neighboring motif and a 40 nt downstream DNA sequence (protospacer). Two complementary DNA oligonucleotides were then designed based on the protospacer and synthesized. The spacer fragments were generated by annealing the corresponding complementary oligonucleotides and inserted into the BspQI site of pGE, producing plasmids carrying the artificial mini-CRISPR arrays. The donor DNA fragment was generated by splicing and overlapping extension (SOE)-PCR with *ApexHF* HS DNA Polymerase FS (Accurate Biotechnology, Hunan, China). The Donor DNA fragments were ligated to the XhoI and SphI cleavage sites of pGE by the TEDA method (Xia et al, 2019), generating the editing plasmid.

Cran1 and its mutants were overexpressed in *Sa. islandicus* using the pSeSD vector (Peng et al, 2012; Peng et al, 2017) and in *E. coli* using the pET22b vector. PCR was used to amplify the target fragment, and overlapping extension PCR was used to obtain the site-specific mutants. The *cran1* knockdown vector was constructed by using tandem repeats having multiple spacers. The spacer sites were chosen to contain 40 bp (TAGATTAGTTTTTCCA-TATCCGTTATAGCCAACGACTACG) immediately after the antisense strand GAAAG (Peng et al, 2017), and the multiple spacers were separated by the repeat sequences. The knockdown plasmid was transferred into in situ gene tagging of *cran1* strain for the detection of Cran1 levels with anti-Flag antibody. The primers used in this study are listed in the Appendix (Appendix Table S2).

## Bright-field microscopy

For bright-field microscopy analysis, 5 µl of cell suspension at the indicated time points were examined under a NIKON TI-E inverted

fluorescence microscope (Nikon, Japan) in Differential interference contrast (DIC) mode.

## Flow cytometry analysis

The cell cycle of synchronized cells of E233S carrying pSeSD and the Cran1 overexpression plasmid was analyzed by flow cytometry using an ImageStreamX MarkII quantitative imaging analysis flow cytometer (Merk Millipore, Germany). Briefly, cells were fixed with ethanol at a final concentration of 70% for at least 12 h at each corresponding time point and stained using propidium iodide (PI) at a final concentration of 50 μg/ml. The data were collected for at least 20,000 cells per sample and analyzed using IDEAS data analysis software.

## Cell cycle synchronization

In situ gene tagging of *cran1* strains was synchronized as previously described (Liu et al, 2021b; Yang et al, 2023). Briefly, cells were first grown aerobically at 75 °C with shaking (145 rpm) in 30 ml of STVU medium. When the $OD_{600}$ reached 0.6–0.8, the cells were transferred into 100 ml STVU medium with an initial estimated $OD_{600}$ of 0.05 and cultivated as above. When the $OD_{600}$ reached 0.15–0.2, acetic acid was added at a final concentration of 6 mM, and the cells were blocked at the G2 phase of the cell cycle after 6 h treatment. Then, the cells were collected by centrifugation at $3000 \times g$ for 10 min at room temperature to remove the acetic acid and washed twice with 0.7% (w/v) sucrose. Finally, the cells were resuspended in 100 ml of pre-warmed STVU medium and cultivated as above for subsequent analysis. For the synchronization of cells carrying the plasmid for Cran1 and mutant overexpression, the ATV medium was used to induce protein expression after the removal of acetic acid. The cell cycle was analyzed using flow cytometry.

## Protein expression and purification

Recombinant Cran1 and mutants with C-terminal His-tag were expressed using *Escherichia coli* BL21 (DE3)-RIL cells. Protein expression was induced during logarithmic growth ($OD_{600} = 0.4$–0.8) by the addition of 0.5 mM isopropyl-thiogalactopyranoside (IPTG) followed by an overnight cultivation at 16 °C. Similarly, proteins were purified from *Sa. islandicus* using Cran1 and its mutant overexpression strains. When the over-expression strains were grown in MTV medium to $OD_{600} = 0.2$, the expression was induced by adding arabinose, and the cultivation continued for 24 h. Cells were collected by centrifugation, and the cell pellet was resuspended in buffer A (50 mM Tris-HCl pH 8.0, 300 mM NaCl, 5% Glycerol). Cells were lysed by sonication, and the cell extract was clarified by centrifugation at $13,000 \times g$ for 20 min at 4 °C. The supernatant was incubated at 70 °C for 20 min (for expression in *E. coli* only), centrifuged at $12,000 \times g$ for 20 min again, and then filtered through a membrane filter (0.45 μm). The samples were loaded on to a Ni-NTA agarose column (Invitrogen) pre-equilibrated with buffer A and eluted with a linear imidazole (40–300 mM imidazole) gradient in buffer A. Fractions were pooled and concentrated to 1 ml and purified further by size exclusion chromatography (SEC) using a Superdex 200 column using the buffer A (50 mM Tris-HCl pH 8.0, 300 mM NaCl, 5% glycerol). Finally, the samples were dialyzed in a storage buffer containing

50 mM Tris-HCl pH 7.4, 100 mM NaCl, 1 mM DTT, 0.1 mM EDTA, and 50% glycerol. Protein concentrations were determined using the Bradford protein assay kit (Beyotime), and the purity was analyzed by 15% SDS-PAGE stained with Coomassie blue.

## Western blotting

The expression levels of Cran1 and cell division proteins in the synchronized cells were analyzed by Western blotting. Around $2 \times 10^8$ cells were collected at the indicated time points for each sample and subjected to SDS-PAGE analysis in a 15% gel. The separated proteins were then transferred onto a PVDF membrane. The specific bands were detected by chemiluminescence using Sparkjade ECL super western blotting detection reagents (Shandong Sparkjade Biotechnology Co., Ltd.) according to the manufacturer's instructions. The primary antibodies against ESCRT-III, ESCRT-III-1, and TBP were produced in rabbit (HuaAn Biotechnology Co., Hangzhou, Zhejiang, China) (Liu et al, 2021b; Liu et al, 2017), and the primary antibodies against Flag tag and His tag were purchased from TransGen Biotech company (Beijing, China). The goat anti-rabbit antibodies (Kermey Biotech, Zhengzhou, China) coupled with peroxidase were used as secondary antibodies.

## Electrophoretic mobility shift assay (EMSA)

The 100 nt FAM-labeled Cran1 promoter sequence (Appendix Table S2) were used as the substrates to determine the binding capacity of aCcr1. The binding assay was performed in a 20 μl reaction mixture containing 10 nM dsDNA, 50 mM Tris-HCl, pH 7.4, 5 mM $MgCl_2$, 20 mM NaCl, 50 μg/mL BSA, 1 mM DTT, 5% glycerol, and different concentrations of purified proteins. The reaction mixture was incubated at 37 °C for 30 min before loaded onto a 12% native polyacrylamide gel. After running in 0.5×TBE, the gel was visualized using an Amersham ImageQuant 800 biomolecular imager (Cytiva).

## DNA cleavage assays

Supercoiled pUC19 plasmid (300 ng) was incubated with 2 μM protein in a final volume of 20 μl in a DNA cleavage buffer (20 mM Tris-HCl pH 7.0, 50 mM KCl, 0.1 mg/ml BSA, and 10 mM divalent metal chloride salt). Reactions were carried out at 70 °C for 30 min, and then the samples were quenched with 5 μl of 0.5 M EDTA, pH 8.0. Samples were analyzed via 1% native agarose gel electrophoresis. Supercoiled and nicked plasmids were extracted using the OMEGA Plasmid Extraction Kit, and linearized plasmids were generated by HindIII single restriction. The signal of the initial DNA substrate (linearized or supercoiled plasmid) was measured and quantified using the ImageJ software if required. The site-directed nuclease and ATPase mutants were assayed under optimal divalent metal conditions of 10 mM $MgCl_2$ and 1 mM $CaCl_2$. The cleavage efficiency was quantified by comparing the band intensity in each lane and calculating the percentage of DNA digested relative to the control.

## ATPase assays

The ATPase activity was examined using a colorimetric malachite green assay that monitored the amount of free phosphate released over

time (Monroe et al, 2014). The reactions in a mixture containing 0.2 μM Cran1 in 25 mM Tris-HCl pH 7.4, 100 mM NaCl, 5 mM MgCl$_2$, and 1 mM ATP in a total volume of 50 μl were carried out for at least 5 min at the indicated temperatures (30, 40, 50, 60, 70, 80, or 90 °C). The reaction mixture was then placed on ice and quenched with 100 μl of malachite green color reagent (14 mM ammonium molybdate, 1.3 M HCl, 0.1% (v/v) Triton X-100, and 1.5 mM malachite green) and 50 μl of 21% (w/v) citric acid. The green compound formed by malachite green, molybdate, and free phosphate was detected by absorbance at 650 nm using a spectrophotometer. A sodium phosphate standard curve was used to estimate the amount of phosphate released during ATP hydrolysis. The NTP hydrolysis activity assays of Cran1 site mutants were performed at 70 °C. It should be noted that above 70 °C, ATP is partially hydrolyzed by heat, so the amount hydrolyzed by heat was excluded from the calculation of ATP hydrolysis activity at 70, 80, and 90 °C.

## Spot assays

For the phage infection assay of *E. coli*, *cran1* sequence or *cran1* with *sire_0087* were cloned into the pET22b generating a vector for the expression of recombinant Cran1. The plasmid was transformed into *E. coli* MG1655. A colony was picked up from the plate and grown in LB broth containing ampicillin (100 mg/ml) at 37 °C to an OD$_{600}$ of 0.4. Protein expression was induced by the addition of 0.2 mM IPTG. After 1 h of growth, 500 μl of bacterial culture was mixed with 10 ml of 0.5% LB top layer agar, and the whole sample was poured onto an LB plate containing ampicillin (100 mg/ml) and IPTG (0.1 mM). Plates were spotted with 3 μl of six tenfold dilutions of phage T1-T7 of 10$^0$–10$^{-5}$ diluted in LB liquid medium. The plates were incubated overnight at 37 °C and then imaged. The spot assay of *Sa. islandicus* REY15A was according to the previous report (Liu et al, 2021a). Briefly, the control (Sis/pSeSD), Cran1, and Cran1 and Sire_0087 co-expression cells were collected at mid-logarithmic phase and mixed with 10 ml of pre-heated ATYS medium containing 0.4% (wt/vol) phytagel. The samples were mixed gently and immediately poured into Petri dishes. After the plates were solidified, 10 μl of the tenfold serially diluted STSV2 or SMV1 preparations were applied to the plates. The plates were incubated at 75 °C for 2–3 days.

## Chromatin fractionation

Chromatin fractionation analyses were similar to those previously described (Takemata et al, 2019). In brief, the cells were harvested from the cultures at OD$_{600}$ = 0.4 for exponential samples and OD$_{600}$ = 1.0 for stationary samples. The cells were resuspended in 100 μl/1 OD$_{600}$ unit/ml of chromatin extraction buffer (25 mM HEPES, 15 mM MgCl$_2$, 100 mM NaCl, 400 mM sorbitol, 0.5% Triton X-100, pH 7.5) and incubated on ice for 10 min. The extracts were centrifuged for 20 min at 14,000× *g*, 4 °C. The soluble fraction was transferred to a new tube, and the pelleted chromatin fraction was resuspended in a volume of chromatin extraction buffer equivalent to the soluble fraction. The chromatin fraction was then sonicated three times for 30 s each.

## Immunoprecipitation and mass spectrometry

In all, 1 L of *Sa. islandicus* culture (Cran1 in situ Flag-tagged strain) in logarithmic growth phase (OD$_{600}$ = 0.3 –0.6) was collected. The cells were suspended in 1 mL of lysis buffer (50 mM Tris-HCl, pH

6.8, 150 mM NaCl, 0.1 mM DTT, 10%, w/v, glycerol, 0.5%) containing protease inhibitor PMSF (Sigma-Aldrich). The cells were broken by ultrasonication. The sample was centrifuged at 20,000×*g* at 4 °C for 30 min, and the supernatant was collected and used as the crude extract (input). Then RNase A was added into 500 μL of crude cell extract. The sample incubated on ice for 30 min to remove RNA. Next, 50 μL anti-FLAG® M2 magnetic beads (Sigma-Aldrich) were added to the sample, and the tube was incubated at 4 °C with shaking for 1 h. The beads were collected by using the magnetic rack and the supernatant (FT, flow through) was discarded. The magnetic beads were washed 5 times with lysis buffer. Finally, the proteins were eluted with 100 μl of 0.1 M glycine HCl, pH 3.0. The eluate was added to 10 μl Tris (1 M), pH 8.0 to protect the eluted proteins from prolonged periods of time in the pH 3.0 solution. The eluate sample was added to 20 μL of 5× loading buffer (100 mM Tris-HCl, pH 6.8, 200 mM β-mercaptoethanol, 4% SDS, 0.2% bromophenol blue, 20% glycerol). After boiling for 10 min, the tube was centrifuged, and the supernatant was subjected to SDS-PAGE. The bands were sliced and subjected to LC-MS analysis at Beijing Protein Innovation Co., Ltd. The identified proteins are listed in Dataset EV2.

## Data availability

The list of sequences used for the phylogeny analysis in this study can be found in Dataset Ev1.

The source data of this paper are collected in the following database record: biostudies:S-SCDT-10_1038-S44319-025-00650-y.

## Peer review information

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

## Acknowledgements

We would like to thank members of the CRISPR and Archaea Biology Research Center for helpful discussions and technicians from the Core Facilities for Life and Environmental Sciences, State Key Laboratory of Microbial Technology of Shandong University for assistance. This work was supported by the National Natural Science Foundation of China [32393973 and 32370033, the State Key Laboratory of Microbial Technology Open Projects Fund (Project No. M2023-20) to YS, and Postdoctoral Fellowship Program of CPSF (GZC20231471) and Postdoc Innovation Project of Shandong Province (SDCX-ZG-202400122) to YY. MK was supported by a grant from the Agence nationale de la recherche (ANR-23-CE13-022).

## Author contributions

**Yunfeng Yang**: Conceptualization; Data curation; Formal analysis; Funding acquisition; Investigation; Methodology; Writing—original draft; Writing—review and editing. **Shikuan Liang**: Data curation; Investigation. **Junfeng Liu**: Formal analysis; Methodology. **Xiaofei Fu**: Investigation. **Pengju Wu**: Formal analysis; Methodology; Writing—review and editing. **Haodun Li**: Methodology; Writing—review and editing. **Jinfeng Ni**: Writing—review and editing. **Qunxin She**: Methodology; Writing—review and editing. **Mart Krupovic**: Formal analysis; Methodology; Writing—review and editing. **Yulong Shen**: Conceptualization; Formal analysis; Supervision; Funding acquisition; Writing—original draft; Project administration; Writing—review and editing.

Source data underlying figure panels in this paper may have individual authorship assigned. Where available, figure panel/source data authorship is listed in the following database record: biostudies:S-SCDT-10_1038-S44319-025-00650-y.

## Disclosure and competing interests statement

The authors declare no competing interests.

# Expanded View Figures

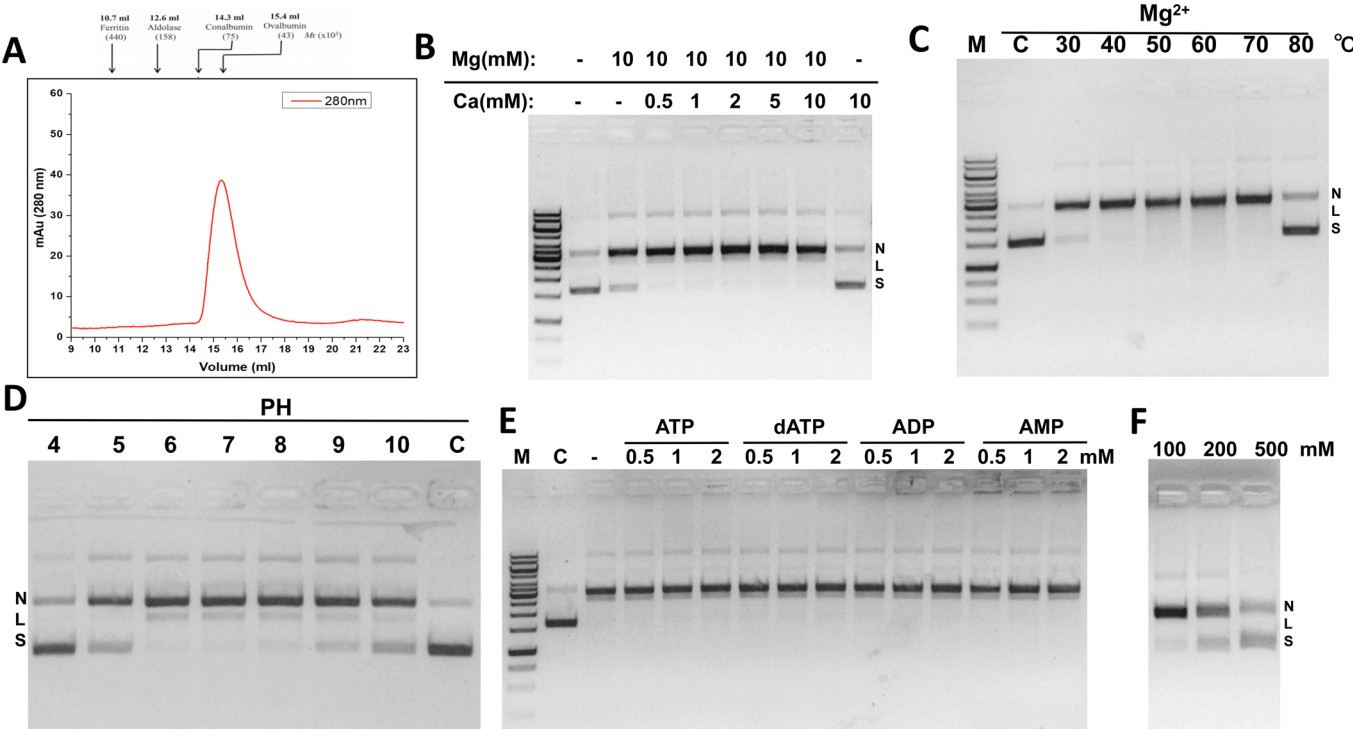

**Figure EV1. Biochemical properties of Cran1 in vitro.**

(**A**) Size exclusion profile of the purified wild-type Cran1. The protein was expressed in *E. coli* BL21 and purified by heat treatment, nickel affinity, and gel filtration with a Superdex 200 column as described in the "Methods". Cran1 was assayed for nickase activity under different temperature gradient (**B**) as well as PH gradient conditions (**C**). C refers to the control reaction without Cran1 at pH 7 and 37 °C. In these reactions, 300 ng of PUC19 DNA was incubated with 2 µM Cran1 in a final volume of 20 µl. Reactions were performed at 70 °C for 30 min and then stopped by the addition of 4 µl of 6×loading dye containing 20 mM EDTA. Samples were analyzed via native agarose gel electrophoresis. (**D**) Gradual increase in the concentration of calcium ions under 10 mM magnesium ions to observe the effect of two-metal ion-catalyzed conditions on the activity of Cran1 nickase. (**E**) Observation of the effect of 0.5, 1, and 2 mM of ATP, dATP, AMP, and ADP on the activity of Cran1 nickase. (**F**) Effect of KCl at 100, 200, and 500 mM on Cran1 activity. 'N', 'L', and 'S' denote the positions of 'nicked', 'linearized', and 'supercoiled' DNA, respectively. Source data are available online for this figure.

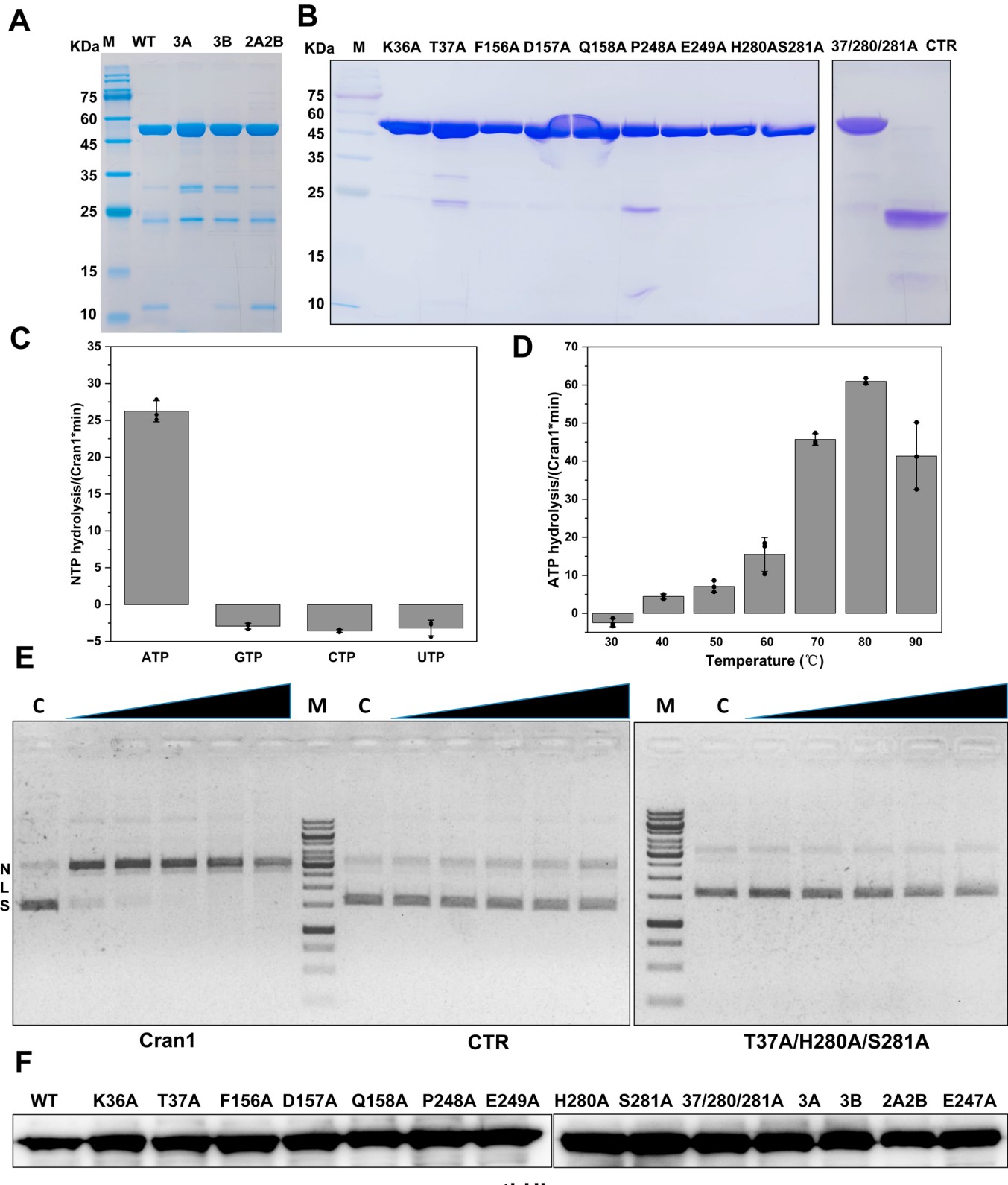

◄

**Figure EV2. Analysis of the activities of the wild-type and the mutant proteins of Cran1.**

(A, B) SDS-PAGE analysis of the wild-type, nuclease active site mutants (A) and the ATPase active site mutants (B) of Cran1. (C) Assay of the ability to hydrolyze ATP, GTP, CTP, or UTP. Data from three independent replicates were shown as mean ± SD. Error bars represent standard deviations from three independent measurements. Source data are available online for this figure. (D) The hydrolytic activity of Cran1 on ATP under different temperature conditions. Data from three independent replicates were shown as mean ± SD. Error bars represent standard deviations from three independent measurements. Source data are available online for this figure. (E) The ATPase activity of Cran1 is essential for the performance of nuclease functions. The cleavage experiments of wild-type Cran1 as well as mutant CTR, T37A/H280A/S281A were carried out under the same conditions as in the previous reaction, with magnesium ion conditions at 70 °C for 30 min, and protein concentration gradients of 0.25, 0.5, 1, 2, and 4 μM. 'N', 'L', and 'S' denote the positions of 'nicked', 'linearized', and 'supercoiled' DNA, respectively. (F) Analysis of the protein levels in cells overexpressing the wild-type Cran1 and its mutant by Western blotting using anti-His tag antibody. Source data are available online for this figure.

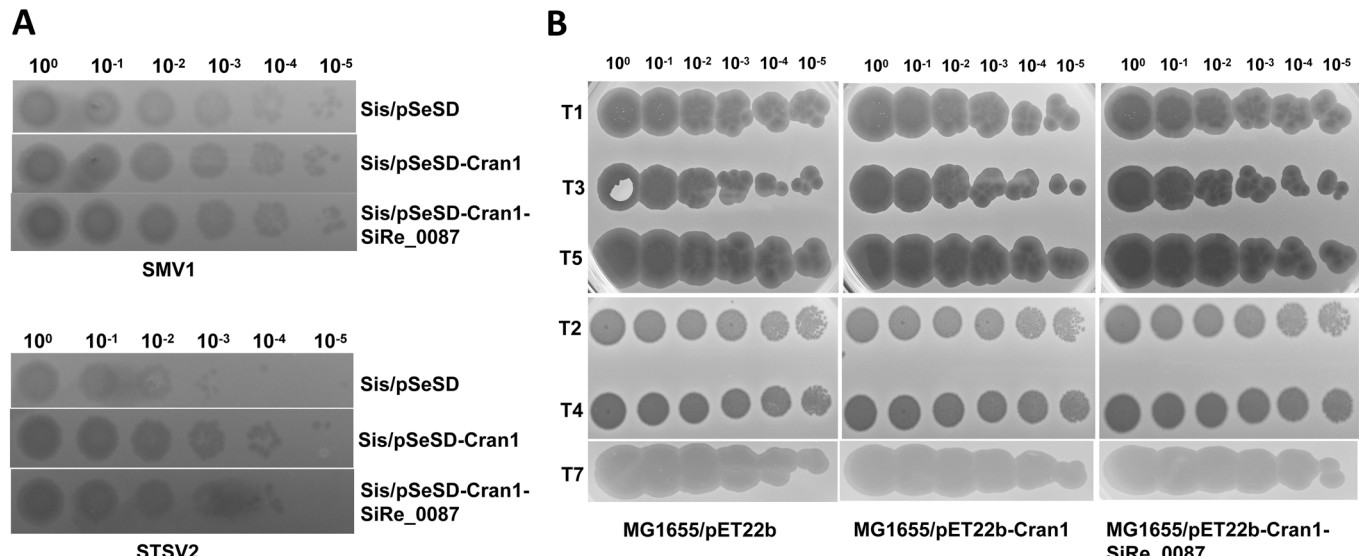

**Figure EV3. Cran1 is not involved in virus defense.**

(A) Spot assay with the *Sa. islandicus* virus SMV1 and STSV2 infecting the cells overexpressing Cran1 and cells co-expressing Cran1 and SiRe_0087. (B) Spot assay of the infection of phage T1, T2, T3, T4, T5, T7 on *Escherichia coli* MG1655 cells expressing Cran1 or co-expressing Cran1 and SiRe_0087. Both archaea virus and bacterial phages were diluted in a tenfold gradient. Cells carrying the empty vectors pSeSD and pET22b were used as controls. Source data are available online for this figure.

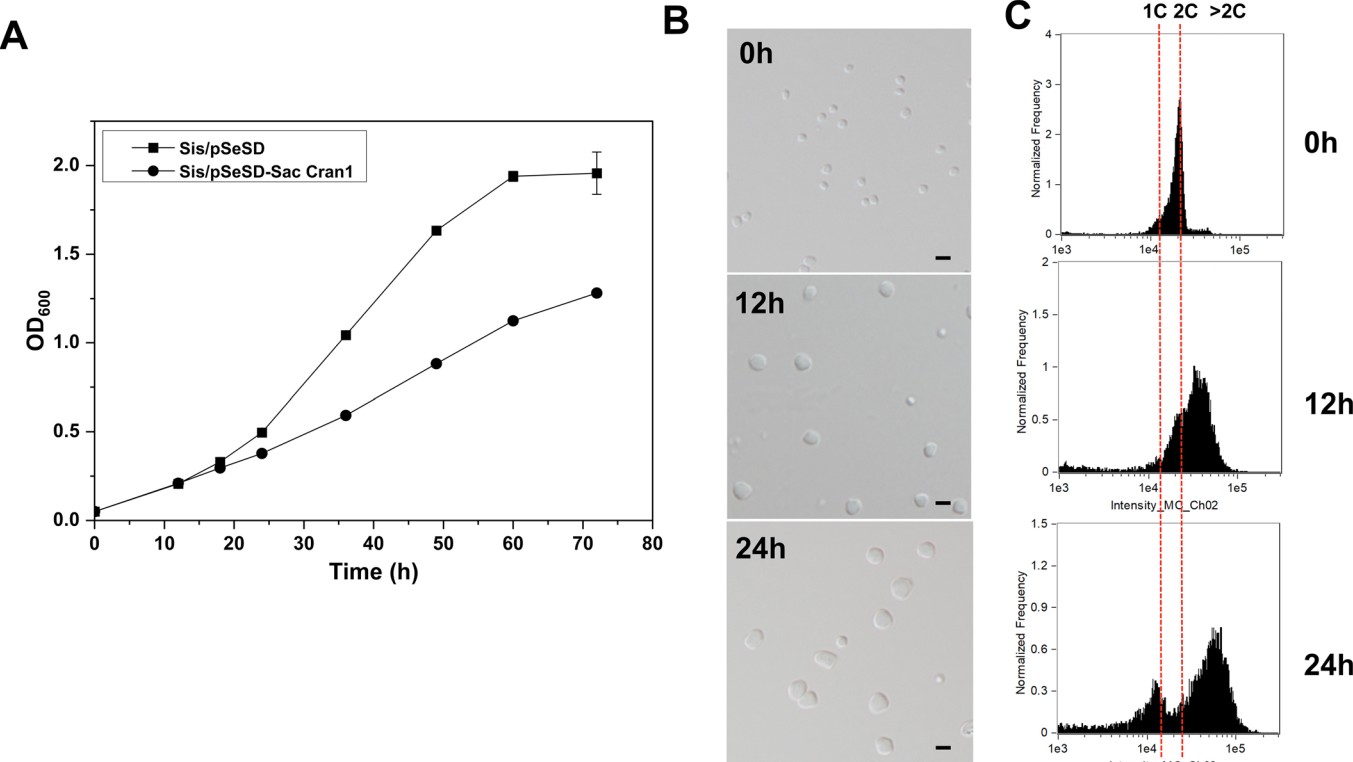

**Figure EV4. Cells with overexpression of the Cran1 homolog from *S. acidocaldarius* (Saci_0157, SacCran1) has similar phenotypes as those of Cran1.**

(**A**) Growth curves of strain Sis/pSeSD-Saci_0157. The cells were inoculated into 30 ml ATV medium to a final estimated $OD_{600}$ of 0.03 and the growth was monitored using spectrometer. Each value was based on data from three independent repeats. Cell harboring the empty plasmid pSeSD was used as a control. (**B**) Differential interference contrast (DIC) mode microscopy and (**C**) flow cytometry of cells overexpressing SacCran1. Cells cultured in the induction medium ATV were taken at different time and observed under an inverted fluorescence microscope. DNA content of cells was analyzed using an ImageStreamX MarkII Quantitative imaging analysis flow cytometry (Merck Millipore, Germany). Scale bars: 2 μm. Source data are available online for this figure.

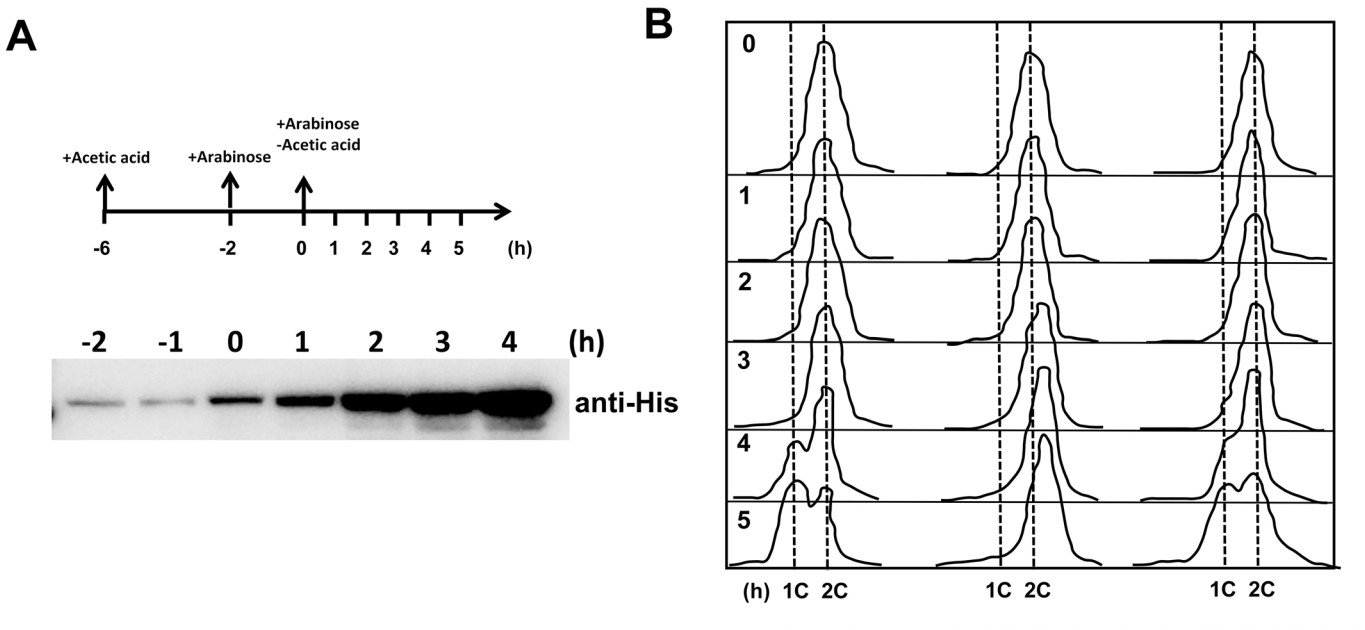

**Figure EV5. The overexpression of Cran1 impairs the cell cycle progression in cells synchronized at the G2 phase.**

(A) A schematic showing cell synchronization and induction of Cran1 overexpression with arabinose (0.2%). The time for the acetic acid treatment and arabinose induction are indicated. E233S containing the empty plasmid (Sis/pSeSD) was used as a control. (B) Flow cytometry profiles of DNA content distribution of cells 0–5 h after acetic acid removal. Source data are available online for this figure.

