## [Peer Review File · EMBO Reports]

Cran1, member of a new class of OLD family ATPases, functions in cell cycle progression in an archaeon

Yunfeng Yang, Shikuan Liang, Junfeng Liu, Xiaofei Fu, Pengju Wu, Haodun Li, Jinfeng Ni, Qunxin She, Mart Krupovic, and Yulong Shen

Corresponding author(s): Yulong Shen (yulgshen@sdu.edu.cn)

Review Timeline:

Submission Date:	1st Apr 25
Editorial Decision:	14th May 25
Revision Received:	17th Aug 25
Editorial Decision:	6th Oct 25
Revision Received:	20th Oct 25
Accepted:	12th Nov 25

Editor: Deniz Senyilmaz Tiebe / Kurt Weir

Transaction Report:

Dear Dr. Shen,

Thank you for transferring your manuscript to EMBO Reports, which was now seen by three referees, whose reports are copied below.

Referees express interest in the proposed role of the OLD family enzyme of *Saccharolobus islandicus* REY15A (SisOLD) in regulation of cell cycle. However, they also raise some concerns that need to be addressed to consider publication here.

I find the reports informed and constructive, and believe that addressing the concerns raised will significantly strengthen the manuscript. As the reports are below, and I think all points need to be addressed, I will not detail them here. Please contact me if you have questions or comments regarding the revision for further discussion (also by video chat).

Given these recommendations, we would like to invite you to revise your manuscript with the understanding that the referee concerns (as in their reports) must be fully addressed and their suggestions taken on board. Please address all referee concerns in a complete point-by-point response. Acceptance of the manuscript will depend on a positive outcome of a second round of review. It is EMBO reports policy to allow a single round of major experimental revision only and acceptance or rejection of the manuscript will therefore depend on the completeness of your responses included in the next, final version of the manuscript.

We realize that it is difficult to revise to a specific deadline. In the interest of protecting the conceptual advance provided by the work, we recommend a revision within 3 months. Please discuss the revision progress ahead of this time with me if you require more time to complete the revisions, or if you have questions or comments regarding the revision (also by video chat).

1. A data availability section providing access to data deposited in public databases is missing (where applicable).
2. Your manuscript contains statistics and error bars based on $n=2$. Please use scatter plots in these cases.

You can submit the revision either as a Scientific Report or as a Research Article. For Scientific Reports, the revised manuscript can contain up to 5 main figures and 5 Expanded View figures, and it should not exceed 27000 characters. If the revision leads to a manuscript with more than 5 main figures it will be published as a Research Article. In this case the Results and Discussion section should be separate. If a Scientific Report is submitted, these sections have to be combined. This will help to shorten the manuscript text by eliminating some redundancy that is inevitable when discussing the same experiments twice. In either case, all materials and methods should be included in the main manuscript file.

4) a .docx formatted letter INCLUDING the reviewers' reports and your detailed point-by-point responses to their comments. As part of the EMBO publication's Transparent Editorial Process, EMBO reports publishes online a Review Process File (RPF) to accompany accepted manuscripts. This File will be published in conjunction with your paper and will include the referee reports, your point-by-point response and all pertinent correspondence relating to the manuscript.

<https://www.embopress.org/page/journal/14693178/authorguide#transparentprocess>

5) a complete author checklist, which you can download from our author guidelines

<https://www.embopress.org/page/journal/14693178/authorguide>. Please insert information in the checklist that is also reflected in the manuscript. The completed author checklist will also be part of the RPF.

6) Please note that all corresponding authors are required to supply an ORCID ID for their name upon submission of a revised manuscript (<<https://orcid.org/>>). Please find instructions on how to link your ORCID ID to your account in our manuscript tracking system in our Author guidelines

<<https://www.embopress.org/page/journal/14693178/authorguide#authorshipguidelines>>

7) Before submitting your revision, primary datasets produced in this study need to be deposited in an appropriate public database (see <https://www.embopress.org/page/journal/14693178/authorguide#datadeposition>). Please remember to provide a reviewer password if the datasets are not yet public. The accession numbers and database should be listed in a formal "Data Availability" section placed after Materials & Method (see also

<https://www.embopress.org/page/journal/14693178/authorguide#datadeposition>). Please note that the Data Availability Section is restricted to new primary data that are part of this study. * Note - All links should resolve to a page where the data can be accessed. *

Additional information on source data and instruction on how to label the files are available:

<https://www.embopress.org/page/journal/14693178/authorguide#sourcedata>

9) Our journal encourages inclusion of *data citations in the reference list* to directly cite datasets that were re-used and obtained from public databases. Data citations in the article text are distinct from normal bibliographical citations and should directly link to the database records from which the data can be accessed. In the main text, data citations are formatted as follows: "Data ref: Smith et al, 2001" or "Data ref: NCBI Sequence Read Archive PRJNA342805, 2017". In the Reference list, data citations must be labeled with "[DATASET]". A data reference must provide the database name, accession number/identifiers and a resolvable link to the landing page from which the data can be accessed at the end of the reference. Further instructions are available at <http://www.embopress.org/page/journal/14693178/authorguide#referencesformat>

10) Regarding data quantification (see Figure Legends:

<https://www.embopress.org/page/journal/14693178/authorguide#figureformat>)

- the name of the statistical test used to generate error bars and P values,
- the number (n) of independent experiments (please specify technical or biological replicates) underlying each data point,
- the nature of the bars and error bars (s.d., s.e.m.),
- If the data are obtained from n Program fragment delivered error `Can't locate object method "less" via package "than" (perhaps you forgot to load "than"?) at //ejpvfs23/sites23b/embor_www/letters/embor_decision_revise_and_review.txt line 56.' 2, use scatter blots showing the individual data points.

12) Please also note our reference format:

13) All Materials and Methods need to be described in the main text using our 'Structured Methods' format, which is required for all research articles. According to this format, the Methods section includes a Reagents and Tools Table (listing key reagents, experimental models, software and relevant equipment and including their sources and relevant identifiers) followed by a Methods and Protocols section describing the methods using a step-by-step protocol format. The aim is to facilitate adoption of the methodologies across labs. More information on how to adhere to this format as well as a downloadable template (.docx) for the Reagents and Tools Table can be found in our author guidelines:

I look forward to seeing a revised version of your manuscript when it is ready. Please let me know if you have questions or comments regarding the revision.

Kind regards,

Deniz Senyilmaz Tiebe

Deniz Senyilmaz Tiebe, PhD
Senior Scientific Editor
EMBO Reports

Referee #1:

Peer review report for Yang et al.

Summary:

In this study, Yang et al. perform a biochemical and cell biological characterization of an archaeal member of the poorly understood OLD nuclease in *S. islandicus*. In their paper, the authors show that sisOLD functions as both ATPase and DNA nickase. Current knowledge of the bacterial OLD family members suggests that they function as part of the phage defence system. By carrying out a phylogenetic analysis and viral infection assays, the authors challenge this notion and show that sisOLD is not likely involved in archaeal immunity against viruses.

Instead, they suggest that sisOLD is a cell cycle regulator. This conclusion is based on two pieces of evidence: sisOLD is expressed in a cell cycle dependent manner, and its overexpression prevents cell division. The authors argue that the ATPase activity of sisOLD is necessary for its function in the cell cycle, while the nickase activity appears dispensable. Lastly, based on some simple pull-down experiments, the authors conclude that sisOLD is likely involved in chromosome remodelling during S-phase to guarantee cell cycle progression.

The text reads well, the experiments have generally been performed to a high standard, the Figures are clear, and Figures 1-3 (and S1-3 and S5) are excellent.

We have a few comments that we recommend the authors address to strengthen their paper prior to publication.

Major points

1. The overexpression of sisOLD clearly induces strong division defects. Most strikingly, it blocks cells leaving an acetic acid arrest from re-entering the following G1. However, little evidence is presented to support the idea that it has a role in S-phase. For example, it seems clear from the data in Fig. 5D that cells overexpressing sisOLD can generate up to 3 or more genomic copies, ruling out a blockage in DNA synthesis. This contradicts the statement in page 19. Because of this, the authors should either revise their assessment or do more work to determine whether or not it has a role in DNA replication.

2. In Figure 4, sisOLD protein expression appears to increase strongly in ~M-phase, implying that OLD may function in the division phase, in addition to or as an alternative to S-phase. This timing fits with the authors previously published observation that OLD is silenced by aCcr1 along with CdvA, and with the accumulation of its *S. aci* homolog in Bortezomib treated cells.

Given this, it would be useful to include a qPCR analysis of synchronised cells to precisely determine more precisely when endogenous OLD is expressed. Any model proposed should take these considerations into account, e.g. by toning down conclusions if there is insufficient evidence to support them.

3. It is hard to reconcile the idea that i) OLD has chromatin remodelling activity and ii) that this plays a role in cell cycle progression, since the mutation experiments show that DNA nicking activity is likely dispensable for its cell cycle phenotype, while the ATP hydrolysis activity is essential. What kind of chromatin remodelling activity were the authors considering?
4. The immunoprecipitation experiment only shows that sisOLD can interact with chromatin or chromatin binding proteins, which is not surprising based on its DNA binding activity as a nickase. Since no direct data is presented to support the idea that there is a chromatin remodelling function besides the nickase activity, we recommend the authors tone down on this conclusion unless additional evidence can be provided. This part of the work might even be removed and included as a part of a future more comprehensive follow up study.
5. Fig. S2 shows that sisOLD can nick the DNA in the absence of ATP, when in the presence of ADP/AMP/dATP. This contradicts with the conclusion from Fig. S3 that ATP hydrolysis is necessary for the nickase activity. The authors should explain this discrepancy.

Minor points

1. Based on the phylogenetic analysis, why don't the authors conclude that sisOLD is part of Clade 3? What might this mean for their analysis? Doesn't this show that Clade 3 is not bacteria specific as stated?
2. It is surprising that extensive mutations are needed to strongly suppress the ATPase activity given that the key motifs in Walker type ATPase like sisOLD is fairly well characterised. Can the authors examine the nucleotide binding pocket with AlphaFold3 in the presence of ATP (assuming that the prediction is with good confidence) to understand if there is any unusual structural feature?
3. Mutating the catalytic glutamate of ATP hydrolysis within Walker B motif typically kills the ATPase activity. Based on the sequence alignment in Fig. S1, this likely corresponds to Glu247 rather than Glu249 mutation used in Fig. 3. Can the authors explain the choice of the mutations in the D-loop, and address the impact of mutating the canonical Walker B motif?
4. The authors state that sisOLD elutes as a monomer, but the buffer in this experiment appears to lack nucleotides. Given that many Walker type ATPases oligomerise in a nucleotide-dependent manner, it is possible that the dimerization of sisOLD requires ATP or ADP. If sisOLD can form dimer in the presence of nucleotide, this raises the possibility that ATPase deficient mutants may inhibit the endogenous sisOLD, which can partially explain the difference in phenotypes observed in Fig. 6 (i.e., gain-of-function vs loss-of-function phenotype). Please clarify.
5. The expression level of the mutant constructs used in Fig. 6 should be verified to ensure that the cell division defects truly correlate with ATPase activity rather than different induction level of the mutant proteins.
6. The knockdown phenotype shown in Fig. S9 is rather mild, likely because of the limited knockdown efficiency. Flow cytometry analysis of knockdown cells stained with both DNA dye and fluorescent ConA might provide a more direct test as to whether the large cells observed in Fig. S9C have more than 2 copies of DNAs as suggested in Fig. S9D.
7. The authors suggest that the 281A mutant might be different from the rest. If this is the case, it is not entirely clear from the image in Figure 6. Can the authors explain why they think the overexpression of the 281A mutant different from the other mutations and, if so, how they explain this? The authors suggest that the overexpression of 281A OLD resembles the effects of Vps4DN over-expression. What is this assessment based on?

Referee #2:

The authors identify a novel cell cycle-related role for OLD family ATPases in Sulfolobales archaea using phylogenetics, biochemistry and genetic perturbations. The paper represents a valuable addition to the archaeal cell cycle literature and the results and methods sections of the paper are cleanly structured and easy to follow.

However I felt the authors could do more to frame possible hypotheses for why an ATPase family involved almost exclusively in antiviral defence would be co-opted for a cell cycle control role in one particular lineage of archaea; moreover, what exactly would this ATPase be doing in the cell cycle? Since the knockdown has only a mild phenotype in contrast to over-expression, this suggests a negative regulatory role, perhaps at the level of chromatin remodelling after replication as the authors suggest. This idea is also not fully fleshed out in the discussion. The discussion is a bit too long as it stands so I would not recommend extending this - rather condensing some sections that are a bit redundant with the results and focussing instead on the broader

outcomes.

Specific comments about the figures and data presentation:

Figure 1 - what is the threshold used to determine the cut-offs for "archaeal", "bacterial", or "mixed"?

Figure 2 - would be better to show the domain structure in the context of other representative OLD family ATPases - same for the alphafold structure in B

Figure 3D - the bars don't need to be different colours - doesn't really need any colour other than black or grey unless the specific residues will be marked in 3A or 3C

Figure 4A - where are the flow cytometry statistics (number of cells, number of independent experiments, etc) reported? Same for Figure 5D

Figure 4B - would be good to quantify the cell-cycle-dependent abundance changes from the western blot (with ≥ 3 biological replicates)

Figure 4A and 4C - confusing use of colours again - the data indicated in red in 4C doesn't match the red in 4A

Figure 6E - the S281A cells have unusual morphologies, is this discussed in the manuscript? I might have missed it

Figure 8 - title - what do the authors refer to in terms of the "cellular amount"?

Referee #3:

Summary:

This manuscript investigates a potentially divergent function of OLD proteins within the archaeon *Saccharolobus islandicus*. The authors use a battery of methods to assay sisOLD function in vitro and in vivo. Already documented to play antiviral roles in bacteria and viruses, the authors claim that the sisOLD enzyme REY15A may play an alternative function in cell cycle. In vitro, they demonstrate that sisOLD carries the typical ATPase and metal-dependent nuclease activities known for OLD proteins, and use structure-guided site directed mutagenesis to characterize the enzyme active sites. In vivo, they show a surprising inability of sisOLD to reduce viral infection. They go on to document the cell cyclic nature of sisOLD protein expression in *S. islandicus*, but sisOLD deletion mutants are not successfully constructed, which the authors use as evidence to claim that it is essential for survival. Cells overexpressing sisOLD bloat to large sizes and do not progress past S phase in 24 hours. The assessment of sisOLD function from many orthogonal approaches is appreciated, especially given the high potential impact for understanding the conservation of cell cycle characteristics between archaea and eukarya. Although the in vitro studies are convincing, the in vivo studies need further experimental support to convincingly claim that sisOLD plays a cell cycle-specific role. Alternatively, the claims need to be substantially softened throughout, and caveats thoroughly explained such that the claims that are sufficiently supported by the data. Specific suggestions for improving the manuscript in these areas are provided below.

Major Points:

1. In the introduction, the second paragraph of the introduction needs to provide a clearer explanation of what is known about the cell cycle of Sulfolobales based on recent literature. Please elaborate a few more sentences explaining the cellular proteins and events that occur stage by stage in the cell cycle of sulfolobales. Then note the similarities / differences to eukarya. This is important because most readers will not be familiar with the cell cycle of archaea as opposed to eukaryotes.
2. What is the cellular / in vivo function of the nuclease domain? It is clearly active as a nuclease in vitro (Fig 2), but it is disappointing that the in vivo role was not addressed in the manuscript (except in the text describing Fig 7, which is rather unclear). Please provide some ideas based on the literature in the discussion.
3. Figure 5, 6, 7. The authors need to provide additional evidence and controls in order to convincingly claim a "Cell cycle defect" in sisOLD overexpression strains by showing that division is actually blocked and mutant cells are truly arrested. Specifically: (a) The doubling time of the wild type expressing the empty vector appears to be about 20 hours (Fig 5A). The flow cytometry and DIC images of sisOLD overexpression strains were taken at 0, 12, and 24 h. This is therefore insufficient time to determine whether the sisOLD overexpressing cells will divide or not. A longer time course is needed and/or live imaging in order to claim a true cell division arrest. (b) Related to this point, appropriate controls are lacking in Fig 5B-D. In addition to the growth curve, please also show DIC images, cell size quantitation, and DNA content analysis of wild type (WT) expressing the empty vector and WT without any vector to confirm that there is no cell shape or division defect that could be caused by the antibiotic itself. (c) In Fig 5A, growth of sisOLD overexpression strain populations, while slow, still continues. Why is cell density increasing if there is a cell division arrest as claimed? This further underscores the need for a longer time course in fig5B-D, and single cell time course imaging. (d) in Fig 7, the authors show flow cytometry data and claim that sisOLD overexpression strains do not divide but they show no microscopy images, movies, or microscopy visualization of stained DNA. This evidence is

needed to directly differentiate the effects of sisOLD overexpression throughout the cell cycle to support their claims. Therefore, an alternative interpretation of the data presented is that sisOLD overexpression could have a size or morphology effect as a result of halted DNA segregation (or increased ploidy, while cell division continues, just with large cells) rather than a cell division effect per se. Can the authors provide more support of their interpretation as opposed to these alternative interpretations (that are equally likely based on the evidence provided)? Or, if these experiments are not possible, please soften the interpretation throughout the manuscript - sisOLD may play a role in cell cycle AND/OR cell morphology / size.

4. The authors need additional genetics evidence to claim that sisOLD is essential. There are many alternative explanations for the failure to isolate a successful knockout, such as: a) off-target issues with suicide plasmid; b) misannotated start or stop codons in the genome for the gene of interest, causing the targeted sequence to disrupt the function of a neighboring essential gene, etc. Instead, the authors should use microbial genetic tools to demonstrate essentiality using these steps: (1) express sisOLD on a plasmid. (2) in that strain, attempt to isolate a knockout strain. Part 2 would be expected to be successful if the gene is essential because the gene of interest is expressed in trans. (3) try to cure the knockout strain of the plasmid by successive transfers in the absence of selection for the plasmid. If plasmid curing is not successful, that is stronger evidence of essentiality than simply failed repeated attempts at knockout. If the suggested experiment is out of scope, please remove the language around "essential", instead stating simply that "knockout mutations were not successful".

5. The evidence for the lack of antiviral activity needs further clarification or experiments to be convincing (Fig S5). Cell survival in the presence of STSV2 appears to be 2 orders of magnitude better than empty vector control during sisOLD overexpression alone or in combination with neighboring gene sire0087. Why do the authors conclude from this that sisOLD is not involved in antiviral defense? My interpretation of these data is that sisOLD indeed protects against viral infection. Can the authors please do a plaque assay in *Sulfolobus* to confirm this result (or maybe that's what the authors did? I don't understand what a "spot assay" is - do they mean "colony forming units through serial dilution plate counts"? This *Sulfolobus* viral infection assay at least needs to be clarified in the methods). Related, can the authors also explain some more context (in introduction and/or discussion) around the specificity of bacterial sisOLD proteins for certain viruses? The example given in the introduction suggests a high degree of specificity (for example, OLD in some bacteria "specifically interferes with phage λ replication", pg 2). If OLD proteins are highly specific to certain viruses, then it is possible that the authors just haven't found the right virus that sisOLD protects against. This should be noted in the discussion as a caveat together with more literature context about the specificity of OLD proteins for certain viruses.

6. Can the authors clarify how they're interpreting the correspondence between ATPase activity mutants and cell cycle / cell morphology phenotypes *in vivo*? Based on what evidence did they expect that reducing ATPase activity of the overexpressed sisOLD through point mutations would lead to normal cell division? Said another way, are they interpreting their data to mean that wild type sisOLD plays a role in promoting cell division by virtue of its ATPase activity? If so, how? What is their model for the mechanism based on the data presented? Please clarify in the results and discussion sections.

7. It is really interesting that sisOLD pulls down other chromatin proteins in coIP experiments, but the authors claim based on this that sisOLD plays a role in chromosome segregation. This is an intriguing speculation, but insufficient evidence is provided for this claim. The authors state this as a hypothesis but there are several other alternative interpretations of the pull-down assay, especially given that the current understanding of the function of the chromatin proteins pulled down (sul7, etc) is incomplete. If it interacts with nucleoid associated proteins, SisOLD could also play a role in DNA damage repair, gene regulation, plasmid segregation, etc etc. I suggest moving the following quote to the discussion, soften it, and provide alternative interpretations: "Given that SisOLD has active ATPase and nuclease domains, we hypothesize that it plays a role in chromatin relaxation and/or chromosome remodeling in cooperation with chromatin proteins, such as Cren7, during the S phase, facilitating replication and subsequent chromosome segregation."

Minor Points:

For the first sentence of the introduction, OLD family proteins are described as "widely distributed in bacteria, archaea, phages, and plasmids." I needed to dig through the references within the direct referenced study for this assertion (reference 1, Dot, E.W., Thomason, L.C. and Chappie, J.S. (2023) Everything OLD is new again: How structural, functional, and bioinformatic advances have redefined a neglected nuclease family) for clarification on how plasmid is defined here. I would either drop the detail about plasmids or add extra an example of OLD family proteins found within plasmids next to the already listed examples of OLD proteins within bacteria, archaea, and viruses.

To help explain the dynamics of cell cycle along with its growth phases, I would write one sentence or two connecting specifically how the cell cycle progressed throughout its exponential phase to stationary phase. It is already very clear from the evidence such as Figure 4 how sisOLD has increased expression during growth like G1 and S, but extra clarification on sisOLD and growth phases would help.

Please rename the protein following archaeal nomenclature - only four letters (for example, "OldS" for protein).

Fig 1 - Provide list of sequences / protein accessions used in the phylogenetic analysis. Spell out aLRT abbreviation in the figure legend.

Throughout, the figure legends provide insufficient information to understand the figure fully. For example, in Fig 2 - please spell out in the legend what each lane in Fig2D represent. "N" "L" and "S" are size markers / positive controls I think but what was the enzyme used? What is the "-" lane? Please spell out everything in the figure for every other figure legend. They were hard to interpret.

Responses to the editor and reviewers

Thank you very much for the favorable evaluation of our manuscript and for the valuable feedback provided by the reviewers. We greatly appreciate the time and effort that went into the thoughtful comments and suggestions on our work.

We have carefully incorporated the suggestions into the revised version of the manuscript. For the requested experiments, we have performed experiments using qPCR to analyze the transcription of sisOLD during the cell cycle, as depicted in Figure 4D in the revised manuscript. Additionally, we have constructed and characterized putative ATPase deficient mutant E246A and E247A, as shown in Figure 3D and 6B, and conducted Western blotting to assess the protein levels of the overexpression strains, as illustrated in Figure EV2F .

Striking, following the suggestion by the reviewer 3, we renamed the protein SisOLD as Cran1 for Cell cycle related ATPase and nickase 1. We believe that the new name would reflect its function as an ATPase domain-regulated nuclease involved in cell cycle regulation.

We hope that the quality of our manuscript has been improved following the inclusion of these experimental results and revision. Please find below a detailed response to each specific point that was raised. Thank you once again for your support and guidance in improving our manuscript.

Referee #1:

In this study, Yang et al. perform a biochemical and cell biological characterization of an archaeal member of the poorly understood OLD nuclease in *S. islandicus*. In their paper, the authors show that sisOLD functions as both ATPase and DNA nickase. Current knowledge of the bacterial OLD family members suggests that they function as part of the phage defence system. By carrying out a phylogenetic analysis and viral infection assays, the authors challenge this notion and show that sisOLD is not likely involved in archaeal immunity against viruses.

Instead, they suggest that sisOLD is a cell cycle regulator. This conclusion is based on two pieces of evidence: sisOLD is expressed in a cell cycle dependent manner, and its overexpression prevents cell division. The authors argue that the ATPase activity of sisOLD is necessary for its function in the cell cycle, while the nickase activity appears dispensable. Lastly, based on some simple pull-down experiments, the authors conclude that sisOLD is likely involved in chromosome remodelling during S-phase to guarantee cell cycle progression.

The text reads well, the experiments have generally been performed to a high standard, the Figures are clear, and Figures 1-3 (and S1-3 and S5) are excellent.

We have a few comments that we recommend the authors address to strengthen their paper

prior to publication.

Response:

Thank you for constructive suggestions. We have carefully reviewed your comments and have made necessary revisions to address the issues raised.

Major points

1. The overexpression of sisOLD clearly induces strong division defects. Most strikingly, it blocks cells leaving an acetic acid arrest from re-entering the following G1. However, little evidence is presented to support the idea that it has a role in S-phase. For example, it seems clear from the data in Fig. 5D that cells overexpressing sisOLD can generate up to 3 or more genomic copies, ruling out a blockage in DNA synthesis. This contradicts the statement in page 19. Because of this, the authors should either revise their assessment or do more work to determine whether or not it has a role in DNA replication.

Response:

Western blot (WB) experiments conducted on synchronized wild-type cells have revealed that the expression of SisOLD is notably elevated during the M-G1 to S phase, indicating a potential role in chromatin processing during the transition from M-G1 to S phase. It is postulated that SisOLD facilitates DNA synthesis rather than inhibiting it, potentially by promoting chromatin relaxation to facilitate DNA replication. We hypothesize that under normal circumstances, SisOLD levels are regulated to be relatively low prior to cell division. Excessive expression of SisOLD may disrupt the coordination between DNA synthesis and cell division, leading to the observed phenotype, that is, enlarged cells with multiple genomic copies. The findings of the study appear to be generally consistent with these hypotheses. For accuracy, we have modified the statements related to the functional role in the revised manuscript. In particular, the subtitle of the last section of the Results on (page 23) has been revised as “SisOLD may be involved in chromatin processing related activities during M-G1 and S phases”.

2. In Figure 4, sisOLD protein expression appears to increase strongly in ~M-phase, implying that OLD may function in the division phase, in addition to or as an alternative to S-phase. This timing fits with the authors previously published observation that OLD is silenced by aCcr1 along with CdvA, and with the accumulation of its S. aci homolog in Bortezomib treated cells. Given this, it would be useful to include a qPCR analysis of synchronized cells to precisely determine more precisely when endogenous OLD is expressed. Any model proposed should take these considerations into account, e.g. by toning down conclusions if there is insufficient evidence to support them.

Response:

Thank you for the thoughtful consideration and valuable suggestions. Following your suggestion, we performed RT-qPCR analysis on synchronized cells to investigate sisOLD transcription. The results, depicted in Figure 4D of the revised manuscript, demonstrate that *sisOLD* expression occurs slightly later than that of *cdvA*, supporting our proposed

model.

In the cell cycle of Sulfolobales, the M-G1 phases make up approximately 10-15% of the entire cell cycle, with the M phase accounting for less than 5%. The protein CdvB is a hallmark of the M phase, exhibiting peak expression at 3 hours following the release of the cell cycle arrest at the G2 phase (i.e., removal of acetic acid). Additionally, the SisOLD protein shows its highest expression at 4-5 hours at the protein level, suggesting its involvement in the M-G1 to S phase transition.

Our recent results (in preparation) suggest that transcription of SisOLD is regulated not only by aCcr1, but also by two other aCcr1 homologs. Hence, the expression timing of SisOLD may not necessarily align with that of *cdvA*.

3. It is hard to reconcile the idea that i) OLD has chromatin remodelling activity and ii) that this plays a role in cell cycle progression, since the mutation experiments show that DNA nicking activity is likely dispensable for its cell cycle phenotype, while the ATP hydrolysis activity is essential. What kind of chromatin remodelling activity were the authors considering?

Response:

At present, the specific chromatin remodeling activity of SisOLD remains hypothetical. Although the overexpression of a DNA nicking activity deficient mutant leads to phenotype similar to that of the wild-type overexpression, *in vitro* experiments indicate that mutation of the key ATPase active site residues results in a loss of the nickase activity. This suggests that the integrity of the ATPase domain is essential for nickase activity to occur, similar to what has been described for other members of the OLD superfamily. The apparent dispensability of the nickase activity may be attributed to the presence of the wild type OLD encoded by the chromosomal gene in the nuclease deficient overexpression strain. We hypothesize that SisOLD potentially fulfils a role akin to that of topoisomerase *in vivo*, relaxing chromatin structure to facilitate processes such as replication and transcription. In the revised version of the manuscript, we toned down and clarified our hypothesis on the chromosome remodeling activity.

4. The immunoprecipitation experiment only shows that sisOLD can interact with chromatin or chromatin binding proteins, which is not surprising based on its DNA binding activity as a nickase. Since no direct data is presented to support the idea that there is a chromatin remodelling function besides the nickase activity, we recommend the authors tone down on this conclusion unless additional evidence can be provided. This part of the work might even be removed and included as a part of a future more comprehensive follow up study.

Response:

We agree with the reviewer and toned down the role of SisOLD in chromatin remodeling in the revised manuscript.

5. Fig. S2 shows that sisOLD can nick the DNA in the absence of ATP, when in the presence of ADP/AMP/dATP. This contradicts with the conclusion from Fig. S3 that ATP hydrolysis is necessary for the nickase activity. The authors should explain this discrepancy.

Response:

We apologize for not being sufficiently clear on this aspect in the original version of the manuscript. The results in Expanded View (EV) Figure 2E indicate that the ATPase domain and, in particular, the nucleotide binding site are necessary for the nickase activity. SisOLD has the nickase activity in the presence of all tested nucleotides with the adenine base, i.e., ATP/ADP/AMP/dATP. However, the absence of the ATPase domain or mutations in the nucleotide binding site lead to the loss of nickase activity. We hypothesize that the ATPase domain of SisOLD regulates the nickase activity, and for this function, binding of nucleotide, either ATP or ADP/AMP/dATP, rather than ATP hydrolysis, is necessary. This is similar to conventional Class 2 OLD proteins, for which the ATP hydrolysis is dispensable for the nuclease activity, but ATP binding is suggested to have a regulatory role, controlling how and when the catalytic C-terminal region interacts with the substrate. The section is revised to clarify the results.

Minor points

1. Based on the phylogenetic analysis, why don't the authors conclude that sisOLD is part of Clade 3? What might this mean for their analysis? Doesn't this show that Clade 3 is not bacteria specific as stated?

Response:

The reasons to define the group including SisOLD as a separate class are twofold. First, there is no bootstrap support for this clade's affinity (aLRT<90) to the group including classes 3 (3a and 3b) and 4. Note that class 4 in our analysis appears to be derived from within class 3. Second, and perhaps more importantly, classes of OLD ATPases (classes 1 through 4) are defined not based on phylogenetic analysis, but rather based on the gene neighborhoods and, implicitly, based on their function (Dot et al., 2023, PMID: 37254295). Given that SisOLD-like genes are associated with the Sulfolobales-specific gene encoding for a conserved membrane protein and do not appear to function in defense, SisOLD orthologs qualify as a separate class of OLD ATPases. Thus, we consider our claims on this subject to be valid.

2. It is surprising that extensive mutations are needed to strongly suppress the ATPase activity given that the key motifs in Walker type ATPase like sisOLD is fairly well characterised. Can the authors examine the nucleotide binding pocket with AlphaFold3 in the presence of ATP (assuming that the prediction is with good confidence) to understand if there is any unusual structural feature?

Response:

Thank you for this suggestion. The model shown in the original version of the manuscript was generated using AlphaFold2 in the apo form (without ATP). We have now used AlphaFold3 to generate a structural model of SisOLD with ATP and magnesium ions (Figure 3C). The residues with atomic distances of <5 Å from the bound ATP molecule are highlighted in red. Comparing the newly predicted model with the former one as well as with the previously reported structures of the OLD family proteins, no unusual structural features were identified in either the magnesium ion or ATP binding pockets. The previous set of mutants was created primarily based on sequence

conservation. From the structural model with the bound ATP, we now recognize that some of the mutated residues are distant from the nucleotide and metal ion, explaining why the corresponding mutations had little effect on the activity. Notably, however, T37 and H280, which had the strongest impact on the activity, are in the active site, consistent with the functional data.

3. Mutating the catalytic glutamate of ATP hydrolysis within Walker B motif typically kills the ATPase activity. Based on the sequence alignment in Fig. S1, this likely corresponds to Glu247 rather than Glu249 mutation used in Fig. 3. Can the authors explain the choice of the mutations in the D-loop, and address the impact of mutating the canonical Walker B motif?

Response:

We chose E249 for mutagenesis analysis because it is also conserved, but in retrospect, we acknowledge that this might not have been the best choice. Following your recommendation, we constructed overexpression strains of E247A, together with E246A, individually. In the predictions shown in Figure 3C, when the interaction distance is limited to 5 Å, the oxygen atoms and carbonyl groups of the side chains of E246 and E247 interact with Mg²⁺ ion. However, when the interaction distance is limited to 4 Å, only T37 interacts with Mg²⁺ ion, and this interaction is direct. We also predicted that the motif LLWQG(68-72) is likely to be a residue related to ATP binding, and we need verify its effect on ATPase subsequently in the future. The remaining residues highlighted in Fig. 3C all interact with ATP, including H280. Based on our experimental results, overexpression of E246A and E247A, similar to wild-type SisOLD, results in growth inhibition and cell enlargement phenotypes. At the same time, *in vitro* ATPase activity tests also showed that compared with wild-type SisOLD protein, E246A and E247A mutants only slightly impaired the ATPase activity. Based on our data, it appears that T37 is the most critical residues for the ATPase activity, with H280 also being important for this activity.

4. The authors state that sisOLD elutes as a monomer, but the buffer in this experiment appears to lack nucleotides. Given that many Walker type ATPases oligomerise in a nucleotide-dependent manner, it is possible that the dimerization of sisOLD requires ATP or ADP. If sisOLD can form dimer in the presence of nucleotide, this raises the possibility that ATPase deficient mutants may inhibit the endogenous sisOLD, which can partially explain the difference in phenotypes observed in Fig. 6 (i.e., gain-of-function vs loss-of-function phenotype). Please clarify.

Response:

We performed gel filtration analysis of SisOLD protein in the presence of ATP (Mole ratio 1:1). However, the protein still eluted with a peak at the putative monomeric position. Furthermore, from the predicted structure, SisOLD lacks the dimerization domain of typical class I OLD protein (Figure 2B). Finally, SisOLD purified from either *Sulfolobus islandicus* or *E. coli* exists in monomeric form. As the ATPase domain of SisOLD could function as a sensor of nucleotide (or dNTP) pool, according to our hypothesis, SisOLD may form dimer or oligomer in the presence dNTP or other nucleotides. It would be interesting experiments in our future investigation.

5. The expression level of the mutant constructs used in Fig. 6 should be verified to ensure that the cell division defects truly correlate with ATPase activity rather than different induction level of the mutant proteins.

Response:

The expression levels of the mutants were analyzed by Western blotting (Figure EV2F). The results showed that the mutant proteins are present at similar levels. Therefore the cell division defects appears to correlate with ATPase activity rather than different induction levels of the mutant proteins.

6. The knockdown phenotype shown in Fig. S9 is rather mild, likely because of the limited knockdown efficiency. Flow cytometry analysis of knockdown cells stained with both DNA dye and fluorescent ConA might provide a more direct test as to whether the large cells observed in Fig. S9C have more than 2 copies of DNAs as suggested in Fig. S9D.

Response:

Indeed, the 50% reduction in SisOLD may not be sufficient to cause a pronounced phenotype, but a delay was observed in the growth curve and smaller population of 1C cells was observed in the knock down strain, implying that decrease in the SisOLD levels has negative impact on DNA synthesis and/or chromosome segregation and cell division. Due to technical limitations, we were unable perform the requested experiments and thus toned down the corresponding statements.

7. The authors suggest that the 281A mutant might be different from the rest. If this is the case, it is not entirely clear from the image in Figure 6. Can the authors explain why they think the overexpression of the 281A mutant different from the other mutations and, if so, how they explain this? The authors suggest that the overexpression of 281A OLD resembles the effects of Vps4DN over-expression. What is this assessment based on?

Response:

Our statement was primarily based on the phenotypical similarity between the SisOLDS281A and the Vps4DN mutants. S281A cells exhibit abnormal morphology, appearing as irregularly enlarged dumbbell-shaped or bud-like cells. By contrast, cells overexpressing the wild-type and other mutant SisOLD proteins are uniformly enlarged and spherical. In the revised manuscript we now clearly state how S281A phenotype is different from the rest of the mutants.

Referee #2:

The authors identify a novel cell cycle-related role for OLD family ATPases in Sulfolobales archaea using phylogenetics, biochemistry and genetic perturbations. The paper represents a valuable addition to the archaeal cell cycle literature and the results and methods sections of the paper are cleanly structured and easy to follow.

However I felt the authors could do more to frame possible hypotheses for why an ATPase family involved almost exclusively in antiviral defence would be co-opted for a cell cycle

control role in one particular lineage of archaea; moreover, what exactly would this ATPase be doing in the cell cycle? Since the knockdown has only a mild phenotype in contrast to over-expression, this suggests a negative regulatory role, perhaps at the level of chromatin remodelling after replication as the authors suggest. This idea is also not fully fleshed out in the discussion. The discussion is a bit too long as it stands so I would not recommend extending this - rather condensing some sections that are a bit redundant with the results and focusing instead on the broader outcomes.

Response:

Thank you for the valuable feedback. We have carefully reviewed your suggestions and made necessary revisions to address the issues raised. In general, we hypothesize that SisOLD is probably involved in chromatin processing related activities during M-G1 and S phases, promoting rather than inhibiting DNA synthesis. The discussion is reorganized to be more focused on the potential function of SisOLD during the cell cycle.

Specific comments about the figures and data presentation:

Figure 1 - what is the threshold used to determine the cut-offs for "archaeal", "bacterial", or "mixed"?

Response:

Clades which included >90% of proteins encoded by organisms of the same domain were considered as bacterial and archaeal, respectively. However, when the clades included >10% of proteins encoded by organisms from the other domain, they were denoted as mixed. This is now mentioned in the revised legend of Figure 1. We now also provide a supplementary table, which lists the sequences in each clade and provides their domain affiliation.

Figure 2 - would be better to show the domain structure in the context of other representative OLD family ATPases - same for the AlphaFold structure in B

Response:

The domain and AlphaFold structures of representative OLD family ATPases have been nicely presented in a recent review (Dot et al., 2022, Everything OLD is new again: How structural, functional, and bioinformatic advances have redefined a neglected nuclease family, Molecular Microbiology). In the revised manuscript, we have shown SisOLD and Thermus structure side by side (not superposed) and cited this reference, as well as added description of comparison of SisOLD with other OLD ATPases in the revised manuscript.

Figure 3D - the bars don't need to be different colors - doesn't really need any color other than black or grey unless the specific residues will be marked in 3A or 3C

Response:

Thanks for the suggestion. We have changed the style to black and grey.

Figure 4A - where are the flow cytometry statistics (number of cells, number of independent experiments, etc) reported? Same for Figure 5D

Response:

The procedure for flow cytometric analysis and the number of cells collected are now described in the 'Methods' section. Figures 4A and 5D show representative flow cytometry profiles. For each time point, the number of cells collected is at least 20,000. This is now indicated in the revised text. For the Figure 4 SisOLD in situ Flag strain synchronisation experiments, we have performed more than three biological replicates with good reproducibility. For the SisOLD overexpression strain in Figure 5, we have even performed multiple times, same as the SisOLD wild-type overexpression results in both Figure 6ABC. All the profile shown in these figures are the representative ones.

Figure 4B - would be good to quantify the cell-cycle-dependent abundance changes from the western blot (with ≥ 3 biological replicates)

Response:

We have quantified the cyclic expression levels of SisOLD by Western blotting based on three experimental replicates and results are now depicted in Figure 4C in the revised manuscript.

Figure 4A and 4C - confusing use of colours again - the data indicated in red in 4C doesn't match the red in 4A

Response:

In the revised manuscript, we have removed colors from Fig. 5A and 5C (former 4A and 4C).

Figure 6E - the S281A cells have unusual morphologies, is this discussed in the manuscript? I might have missed it

Response:

We have described the phenotype in section 'The ATPase activity of SisOLD is vital for the cell cycle progression'. S281A exhibits irregular enlargement and significant changes in cell morphology, but we currently do not know the cause of this phenotype, as stated in the revised manuscript.

Figure 8 - title - what do the authors refer to in terms of the "cellular amount"?

Response:

Fig. 8 legend title has been changed to "Subcellular localization of SisOLD."

Referee #3:

Summary:

This manuscript investigates a potentially divergent function of OLD proteins within the archaeon *Saccharolobus islandicus*. The authors use a battery of methods to assay sisOLD function in vitro and in vivo. Already documented to play antiviral roles in bacteria and viruses, the authors claim that the sisOLD enzyme REY15A may play an alternative function in cell cycle. In vitro, they demonstrate that sisOLD carries the typical ATPase and metal-dependent nuclease activities known for OLD proteins, and use structure-guided site directed

mutagenesis to characterize the enzyme active sites. In vivo, they show a surprising inability of sisOLD to reduce viral infection. They go on to document the cell cyclic nature of sisOLD protein expression in *S. islandicus*, but sisOLD deletion mutants are not successfully constructed, which the authors use as evidence to claim that it is essential for survival. Cells overexpressing sisOLD bloat to large sizes and do not progress past S phase in 24 hours. The assessment of sisOLD function from many orthogonal approaches is appreciated, especially given the high potential impact for understanding the conservation of cell cycle characteristics between archaea and eukarya. Although the in vitro studies are convincing, the in vivo studies need further experimental support to convincingly claim that sisOLD plays a cell cycle-specific role. Alternatively, the claims need to be substantially softened throughout, and caveats thoroughly explained such that the claims that are sufficiently supported by the data. Specific suggestions for improving the manuscript in these areas are provided below.

Response:

Thank you for the suggestions. We have carefully reviewed your comments and recommendations, and have made necessary revisions to address the issues raised.

Major Points:

1. In the introduction, the second paragraph of the introduction needs to provide a clearer explanation of what is known about the cell cycle of Sulfolobales based on recent literature. Please elaborate a few more sentences explaining the cellular proteins and events that occur stage by stage in the cell cycle of sulfolobales. Then note the similarities / differences to eukarya. This is important because most readers will not be familiar with the cell cycle of archaea as opposed to eukaryotes.

Response:

We have revised the second paragraph of the Introduction, providing additional description of the events that occur during different cell cycle phases of Sulfolobales.

2. What is the cellular / in vivo function of the nuclease domain? It is clearly active as a nuclease in vitro (Fig 2), but it is disappointing that the in vivo role was not addressed in the manuscript (except in the text describing Fig 7, which is rather unclear). Please provide some ideas based on the literature in the discussion.

Response:

The in vivo function of the nuclease domain remains unclear. The nuclease domain in the other studied OLD family enzymes functions in degradation or cleavage of invading viral or plasmid DNA (Cheng R et al., 2021, A nucleotide-sensing endonuclease from the Gabija bacterial defense system. *Nucleic Acids Res* 49: 5216-5229; Schiltz CJ et al., 2020, The full-length structure of OLD defines the ATP hydrolysis properties and catalytic mechanism of Class 1 OLD family nucleases. *Nucleic Acids Res* 48: 2762-2776). Notably, the nuclease domain of OLD enzymes belongs to the Toprim superfamily, which is also found in topoisomerases, a group of enzymes that resolve topological problems in DNA during various various cellular processes, such as genome replication (Vic Norris et al., 2023, The roles of nucleoid-associated proteins and topoisomerases in chromosome structure, strand segregation, and the generation of phenotypic heterogeneity in

bacteria. *FEMS Microbiology Reviews* 47, 1–22). We hypothesize that the nuclease activity of SisOLD participates in chromatin relaxation, although the detailed mechanism needs further investigation. The lack of phenotype when overexpressing the nuclease-negative versions of SisOLD suggests that the native, chromosomally-encoded SisOLD is sufficient to perform the intended function. This hypothesis is now discussed in the revised manuscript.

3. Figure 5, 6, 7. The authors need to provide additional evidence and controls in order to convincingly claim a "Cell cycle defect" in sisOLD overexpression strains by showing that division is actually blocked and mutant cells are truly arrested. Specifically: (a) The doubling time of the wild type expressing the empty vector appears to be about 20 hours (Fig 5A). The flow cytometry and DIC images of sisOLD overexpression strains were taken at 0, 12, and 24 h. This is therefore insufficient time to determine whether the sisOLD overexpressing cells will divide or not. A longer time course is needed and/or live imaging in order to claim a true cell division arrest.

Response:

The doubling time of actively dividing cell of *Sa. islandicus* is 6h. Thus, sampling time of 12h and the 24h of the overexpression strain appears to be fully sufficient to observe cytokinesis. Moreover, in the flow cytometry profiles at the 24h time point, we can clearly see that the DNA content at this time is much higher than 2C (or even 4C), which indicates that the cells have gone through at least two rounds of genome replication (S phase). Indeed, it would be useful to perform the live-cell imaging experiment, but because Sulfolobales cells grow at 75°C, it requires special equipment, which unfortunately, we do not possess and thus cannot perform this experiment.

(b) Related to this point, appropriate controls are lacking in Fig 5B–D. In addition to the growth curve, please also show DIC images, cell size quantitation, and DNA content analysis of wild type (WT) expressing the empty vector and WT without any vector to confirm that there is no cell shape or division defect that could be caused by the antibiotic itself.

Response:

The overexpression vector pSeSD for *Sa. islandicus* carries a uracil synthesis cassette pyrEF. Therefore, cells harboring the pSeSD vector can grow without addition of uracil, and the selection is not based on bacterial antibiotics (Peng, N., et al., 2012, A Synthetic Arabinose-Inducible Promoter Confers High Levels of Recombinant Protein Expression in Hyperthermophilic Archaeon. *Appl Environ Microb*, 78, 5630-5637). In addition, it has been established that the wild-type *Sa. islandicus* without the plasmid exhibits the same growth characteristics and cell morphology as those carrying the empty plasmid pSeSD (Liu JF, Gao RX, Li CT, Ni JF, Yang ZJ, Zhang Q, Chen HN, Shen YL, 2017, Functional assignment of multiple ESCRT-III homologs in cell division and budding. *Mol Microbiol* 105: 540-553). Further, if cells carrying the SisOLD overexpression vector have not been induced by arabinose, they also exhibit the same phenotype as the wild-type strain. Therefore, we only present the cell morphology and flow cytometry of the SisOLD overexpression strain at 0h because all control cells are identical to those at 0h.

Figure for referee with unpublished data has been removed upon request by the authors.

(c) In Fig 5A, growth of sisOLD overexpression strain populations, while slow, still continues. Why is cell density increasing if there is a cell division arrest as claimed? This further underscores the need for a longer time course in fig5B-D, and single cell time course imaging.

Response:

In Figure 5A, the OD₆₀₀ of the SisOLD-overexpressing strain continues to increase because, although cell division is inhibited, DNA continues to replicate, leading to cell enlargement and increased optical density. It is important to note that after prolonged incubation, arabinose is depleted and SisOLD induction is insufficient, and cells start to divide again. This is now explained in the revised manuscript.

(d) in Fig 7, the authors show flow cytometry data and claim that sisOLD overexpression strains do not divide but they show no microscopy images, movies, or microscopy visualization of stained DNA. This evidence is needed to directly differentiate the effects of sisOLD overexpression throughout the cell cycle to support their claims. Therefore, an alternative interpretation of the data presented is that sisOLD overexpression could have a size or morphology effect as a result of halted DNA segregation (or increased ploidy, while cell division continues, just with large cells) rather than a cell division effect per se. Can the authors provide more support of their interpretation as opposed to these alternative interpretations (that are equally likely based on the evidence provided)? Or, if these experiments are not possible, please soften the interpretation throughout the manuscript - sisOLD may play a role in cell cycle AND/OR cell morphology / size.

Response:

We agree that single-cell imaging would be very useful here. Unfortunately, this technology is currently not available in our lab. However, from the flow cytometry results, it is clearly observed that in the control strain and the strain overexpressing the ATPase-deficient mutant (T37A/H280A/S281A), a 1C peak appears at 4 hours, indicating that cell division has occurred, producing cells with a single genome copy. By contrast, the strain overexpressing the wild-type SisOLD cannot produce 1C cells, indicating that cell division is blocked. Thus, we believe that overexpression of SisOLD blocks cell division. Nevertheless, we have toned down the interpretation in the revised manuscript, as suggested.

4. The authors need additional genetics evidence to claim that sisOLD is essential. There are many alternative explanations for the failure to isolate a successful knockout, such as: a) off-

target issues with suicide plasmid; b) misannotated start or stop codons in the genome for the gene of interest, causing the targeted sequence to disrupt the function of a neighboring essential gene, etc. Instead, the authors should use microbial genetic tools to demonstrate essentiality using these steps: (1) express sisOLD on a plasmid. (2) in that strain, attempt to isolate a knockout strain. Part 2 would be expected to be successful if the gene is essential because the gene of interest is expressed in trans. (3) try to cure the knockout strain of the plasmid by successive transfers in the absence of selection for the plasmid. If plasmid curing is not successful, that is stronger evidence of essentiality than simply failed repeated attempts at knockout. If the suggested experiment is out of scope, please remove the language around "essential", instead stating simply that "knockout mutations were not successful".

Response:

Genetic manipulation in *Sa. Islandicus* REY15A is based on the endogenous CRISPR system, and there have been numerous reports that this editing system is efficient and powerful. If the gene is essential, the genome cannot be rescued by homologous recombination and the transformants cannot be obtained. Therefore, we consider the gene to be essential. Due to the uracil screening markers used for both the overexpression plasmid and the knockout plasmid, it was not possible to perform the suggested experiment. Note that genetic toolkit is very limited in Sulfolobales and we simply do not have a different selection marker at the moment. As suggested by reviewer, we now refrain from claiming "essentiality" and instead state that "knockout mutations were not successful".

5. The evidence for the lack of antiviral activity needs further clarification or experiments to be convincing (Fig S5). Cell survival in the presence of STSV2 appears to be 2 orders of magnitude better than empty vector control during sisOLD overexpression alone or in combination with neighboring gene sire0087. Why do the authors conclude from this that sisOLD is not involved in antiviral defense? My interpretation of these data is that sisOLD indeed protects against viral infection. Can the authors please do a plaque assay in Sulfolobus to confirm this result (or maybe that's what the authors did? I don't understand what a "spot assay" is - do they mean "colony forming units through serial dilution plate counts"? This Sulfolobus viral infection assay at least needs be clarified in the methods). Related, can the authors also explain some more context (in introduction and/or discussion) around the specificity of bacterial sisOLD proteins for certain viruses? The example given in the introduction suggests a high degree of specificity (for example, OLD in soem bacteria "specifically interferes with phage λ replication", pg 2). If OLD proteins are highly specific to certain viruses, then it is possible that the authors just haven't found the right virus that sisOLD protects against. This should be noted in the discussion as a caveat together with more literature context about the specificity of OLD proteins for certain viruses.

Response:

The experiment presented in Figure S5 is a spot assay for archaeal viruses and bacteriophages, not a cell viability assay. Briefly, serial dilutions of virus preparation are spotted on the lawns of the host strains either expressing or not the genes of interest (SisOLD, in our case). The clearance zones represent either cell lysis or inhibition of cell growth. We have added the experimental steps for the *Sa. islandicus* REY15A spot assay

to the Methods section to provide a better understanding of the experimental process. The OLD defence system in bacteria indeed has a defensive effect against specific viruses. However, due to the limited availability of archaeal viruses in our laboratory, we cannot rule out the possibility that we have not found the specific virus targeted by SisOLD. This caveat is now mentioned in the revised text, as suggested.

6. Can the authors clarify how they're interpreting the correspondence between ATPase activity mutants and cell cycle / cell morphology phenotypes in vivo? Based on what evidence did they expect that reducing ATPase activity of the overexpressed sisOLD through point mutations would lead to normal cell division? Said another way, are they interpreting their data to mean that wild type sisOLD plays a role in promoting cell division by virtue of its ATPase activity? If so, how? What is their model for the mechanism based on the data presented? Please clarify in the results and discussion sections.

Response:

Based on our data, we hypothesize that SisOLD promotes DNA replication by participating in a certain chromatin processing process, which remains unclear. This is based on the fact that SisOLD is not expressed before cell division, but begins to be expressed in the late stage of cell division and through G1 and S phases. Overexpression of the wild-type SisOLD appears to over-stimulate DNA synthesis (possibly by hampering chromatin condensation following replication) (Figure 7B). The ATPase-defective SisOLD (more likely in nucleotide binding rather than ATP hydrolysis loses its normal function, and overexpression of the mutant proteins no longer hampers chromatin condensation nor does it inhibit cell division. We have modified the Results and Discussion sections to clarify the interpretation.

7. It is really interesting that sisOLD pulls down other chromatin proteins in colP experiments, but the authors claim based on this that sisOLD plays a role in chromosome segregation. This is an intriguing speculation, but insufficient evidence is provided for this claim. The authors state this as a hypothesis but there are several other alternative interpretations of the pull-down assay, especially given that the current understanding of the function of the chromatin proteins pulled down (sul7, etc) is incomplete. If it interacts with nucleoid associated proteins, SisOLD could also play a role in DNA damage repair, gene regulation, plasmid segregation, etc etc. I suggest moving the following quote to the discussion, soften it, and provide alternative interpretations: "Given that SisOLD has active ATPase and nuclease domains, we hypothesize that it plays a role in chromatin relaxation and/or chromosome remodeling in cooperation with chromatin proteins, such as Cren7, during the S phase, facilitating replication and subsequent chromosome segregation."

Response:

Thank you for your insightful suggestions. We have modified the text, as suggested.

Minor Points:

For the first sentence of the introduction, OLD family proteins are described as "widely distributed in bacteria, archaea, phages, and plasmids." I needed to dig through the

references within the direct referenced study for this assertion (reference 1, Dot, E.W., Thomason, L.C. and Chappie, J.S. (2023) Everything OLD is new again: How structural, functional, and bioinformatic advances have redefined a neglected nuclease family) for clarification on how plasmid is defined here. I would either drop the detail about plasmids or add extra an example of OLD family proteins found within plasmids next to the already listed examples of OLD proteins within bacteria, archaea, and viruses.

Response:

Following your suggested, we have deleted the mention of the plasmids.

To help explain the dynamics of cell cycle along with its growth phases, I would write one sentence or two connecting specifically how the cell cycle progressed throughout its exponential phase to stationary phase. It is already very clear from the evidence such as Figure 4 how sisOLD has increased expression during growth like G1 and S, but extra clarification on sisOLD and growth phases would help.

Response:

The genome replicates quickly during the exponential phase, but due to limitation in available nutrients, the replication during the stationary phase is much slower than during the exponential phase. Thus, the expression level of SisOLD during the exponential phase is also much higher than during the stationary phase. This is now clarified in the revised manuscript.

Please rename the protein following archaeal nomenclature - only four letters (for example, "OldS" for protein).

Response:

Thanks for your constructive suggestion. We have renamed the protein as Cran1 for Cell cycle related ATPase and nickase 1. We believe that the new name would reflect its function as an ATPase domain-regulated nuclease invoved in cell cycle regulation.

Fig 1 - Provide list of sequences / protein accessions used in the phylogenetic analysis. Spell out aLRT abbreviation in the figure legend.

Response:

We now provide a supplementary table, which lists the membership of each clade and provides the accession numbers of the corresponding protein sequences. SH-aLRT (Shimodaira–Hasegawa approximate likelihood ratio test) abbreviation is now also spelled out in the figure legend.

Throughout, the figure legends provide insufficient information to understand the figure fully. For example, in Fig 2 - please spell out in the legend what each lane in Fig2D represent. "N" "L" and "S" are size markers / positive controls I think but what was the enzyme used? What is the "-" lane? Please spell out everything in the figure for every other figure legend. They were hard to interpret.

Response:

We have added more detailed explanations in the figure legends as well as in the Methods section for clarification.

Dear Prof. Shen

Thank you for submitting your revised manuscript. It has now been seen by two of the original referees.

As you will see, referees find that the study is significantly improved during revision and recommend publication. However, referee #1 has a few minor remaining concerns: they request a thorough check for linguistic errors and would like a mention that delayed entry into S phase would increase the number of cells in G1. Please address all concerns textually per referee recommendations. Please provide a point-by-point response. Please let me know if you would like to discuss any of the points further.

Moreover, the editorial points below need to be addressed before I can accept the manuscript.

- Please note that figures should not be included in the manuscript file, only the legends of main and EV figures should remain at the end of the manuscript
- Please reduce the number of keywords on the abstract page to five (ideally choosing broad general terms).
- As we are switching from a free-text author contribution statement towards a more formal statement based on Contributor Role Taxonomy (CRediT) terms, please remove the present Author Contribution section and instead specify each author's contribution(s) directly in the Author Information page of our submission system during upload of the final manuscript. See <https://casrai.org/credit/> for more information.
- All main and EV figures need to be uploaded as individual files, with one file to each figure, with sufficient resolution/quality for production.
- Figure 7A is never called out. Please add an in-text callout for this panel
- Please note that the specific URL for the dataset referenced in Fig 1A should be provided in the data availability statement, i.e., we need a link that resolves directly to the dataset.
- Please note that the data availability statement should only address data in publicly available databases. Any references to author-sharing or supplementary information should be removed.
- Appendix figures should not be uploaded individually, but only as part of a single Appendix PDF. Their figure legends should be removed from the main text and shown directly under each figure in the Appendix, for easy reference.
- Appendix Table S3 and S4 should be converted/renamed (both in the file and in the text) to EXPANDED VIEW DATASETS (call-out: "Dataset EV1/2"), their legends should be removed from the Appendix file, and the spreadsheets each need a separate "Legend" tab containing dataset title and legend information
- Please note that EMBO press papers are accompanied online by:
 - A) a short (2 sentences) summary of the findings and their significance,
 - B) 2-5 short bullet points highlighting the key results, and
 - C) a synopsis image in .jpg or .png format that is exactly 550 pixels wide and 300-600 pixels high (the height is variable). Please note that the text needs to be legible at the final size. Please upload this information along with your revised manuscript (the text for A and B should be provided in one separate Word file uploaded as Synopsis and Bullet Points).
- The Reagent and Tools table you uploaded as an individual file is sufficient. Please remove the Reagent and Tools Table from the manuscript file. For more information, please check <https://www.embopress.org/page/journal/14602075/authorguide#structuredmethods>
- Thanks for providing the source data (SD). Please upload the SD file folders, which are currently all combined in a single ZIP archive, as one separate archive per each main figure. Source data files need to be saved in a scheme one figure/folder with subfolders for each panel and then uploaded as .zip files. E.g. all the Source data files for figure 1 need to be saved in a single folder and this needs to be zipped and then uploaded as "SD figure 1.zip" file. For EV and/or appendix figures, ZIP together all source data.
- Section order should be corrected: Title page - Abstract & Keywords - Introduction - Results - Discussion - Methods - Data Availability - Acknowledgments - Disclosure Statement & Competing Interests - References - Figure Legends - (Main Tables with legends if applicable) - Expanded View Figure Legends.
- Please provide display individual data points in all figures that include $n < 5$
 - o For instance: 3D; 4C; 6A,B,C; EV2C,D
- Our production/data editors have asked you to clarify several points in the figure legends:
 - o Please note that information related to n is missing in the legends of figures 3D, 6A-C; EV2 C, D
 - o Please note that the error bars are not defined in the legends of figures 3D, 4C, D; 5A, 6A-C; EV2 C, D
 - o Please note that the red arrows are not defined in the legend of figure 6E, EV4 C. This needs to be rectified.

Kind regards,
Kurt Weir
Editor
EMBO Reports

Referee #1:

We would like to congratulate the authors.
The revised paper by Yang et al., is much improved.
The new data added strengthen the story.
In our view, it now addresses all the main questions raised in reviews.

A few minor things:

Before publication, we would recommend that the new text be checked carefully for linguistic errors. A few were introduced.

In addition, when discussing the knockdown (which should probably be in the main Figures) the authors should note that a delay in S-phase entry would, if anything, lead to an increase in the G1 peak, not a decrease. A loss of G1 cells typically occurs when division is inhibited entry into S-phase is accelerated.

Finally, in future work it would be better to have more cells in the flow plots to smooth the distributions.

Referee #2:

The authors have satisfactorily addressed my comments, and to the best of my understanding, those of the other reviewers. I recommend publication.

Responses to the referees:

Referee #1:

We would like to congratulate the authors.

The revised paper by Yang et al., is much improved.

The new data added strengthen the story.

In our view, it now addresses all the main questions raised in reviews.

A few minor things:

Before publication, we would recommend that the new text be checked carefully for linguistic errors. A few were introduced.

Response:

Thank you for reviewing our manuscript. We have carefully checked the language, and all the errors identified were corrected.

In addition, when discussing the knockdown (which should probably be in the main Figures) the authors should note that a delay in S-phase entry would, if anything, lead to an increase in the G1 peak, not a decrease. A loss of G1 cells typically occurs when division is inhibited entry into S-phase is accelerated.

Response:

According to the reviewer's suggestion, we have moved the knockdown figure to main figures as Fig. 5.

We appreciate the insightful and helpful advice regarding explanation of increase and loss of G1 cells in the analysis,.

Finally, in future work it would be better to have more cells in the flow plots to smooth the distributions.

Response:

Thanks for the important advice. We will have more cells for flow analysis in our future study.

Referee #2:

The authors have satisfactorily addressed my comments, and to the best of my understanding, those of the other reviewers. I recommend publication.

Response:

Many thanks.

Prof. Yulong Shen
Shandong University
State Key Laboratory of Microbial Technology, School of Life Sciences
Qindao, Shangdong 266237
China

Dear Prof. Shen,

I am pleased to inform you that your manuscript has been accepted for publication in EMBO reports. Your manuscript will be processed for publication by EMBO Press. It will be copy edited and you will receive page proofs prior to publication. Please note that you will be contacted by Springer Nature Author Services to complete licensing and payment information.

Yours sincerely,

Kurt Weir
Editor
EMBO Reports
